# RL, BUT DON'T DO ANYTHING I WOULDN'T DO

## ABSTRACT

In reinforcement learning, if the agent's reward differs from the designers' true utility, even only rarely, the state distribution resulting from the agent's policy can be very bad, in theory and in practice. When RL policies would devolve into undesired behavior, a common countermeasure is KL regularization to a trusted policy ("Don't do anything I wouldn't do"). All current cutting-edge language models are RL agents that are KL-regularized to a "base policy" that is purely predictive. Unfortunately, we demonstrate that when this base policy is a Bayesian predictive model of a trusted policy, the KL constraint is no longer reliable for controlling the behavior of an advanced RL agent. We demonstrate this theoretically using algorithmic information theory, and while systems today are too weak to exhibit this theorized failure precisely, we RL-finetune a language model and find evidence that our formal results are plausibly relevant in practice. We also propose a theoretical alternative that avoids this problem by replacing the "Don't do anything I wouldn't do" principle with "Don't do anything I mightn't do".

## 1 INTRODUCTION

Agents optimizing their objective in a way not intended by designers could be amusing, annoying, insidious, or disastrous. Amusingly, RL researchers attempted to get a simulated humanoid to walk, but the reward resulted in crazy locomotion (Lee et al., 2021). Annoyingly, maximizing a simulated-environment's reward can produce a policy that would achieve little real-world-reward by exploiting errors in the simulation (Mishra et al., 2017; Baker et al., 2019). Insidiously, artificial agents selecting links to maximize click-through on social media sites have succeeded, but also affecting people in ways designers never sought to (Chan et al., 2023). For a much longer list of such failures occurring "in the wild", see (Krakovna, 2018). Finally, sufficiently capable reinforcement learners would likely recognize an incentive to escape human oversight, intervene in the protocol determining their reward, and use force to ensure they can retain control of their reward, subject to such an outcome being possible from the agent's action space, and several other assumptions laid out by Cohen et al. (2022b).

Indeed, several sources suggest that extremely successful reward-maximization is *itself* a sign of bad outcomes for humanity. Zhuang & Hadfield-Menell (2020) demonstrate that in a resource-constrained world, optimizing the world's state to maximize a function of *some* features would, in plausible settings, be arbitrarily bad with respect to a utility function that also cares about *unincluded* features. Turner et al. (2021) develop a formal model of "power"—being able to accomplish a randomly sampled goal—and find that (reward-)optimal policies tend to seek power. And Cohen et al. (2022b) observe that any behavior that ensures that long-term reward is nearly-certainly-maximal must include extensive control over threats to its physical integrity, including threats from humans.

An appealing and popular proposal to avoid such outcomes is to constrain the agent to follow a policy that is not too dissimilar to a more familiar "base policy". This is the approach taken when RL-finetuning large language models (LLMs). This class of approaches limits the upside of RL, since it forgoes optimal policies, but it is a reasonable attempt to avoid catastrophic policies. The KL divergence, in particular KL(proposed policy‖base policy), enforces proximity in a robust, "safety-conscious" way: if basepolicy(action) $\ll 1$ while proposedpolicy(action) $\not\ll 1$, the KL penalty is high, even while $L_p$ norms can be small. For any very bad outcomes that are unlikely under the base policy, this method ensures they remain very unlikely. However, if we ensure that KL(proposed policy‖base policy) is small, but the base policy only *approximates* a trusted policy, to what extent can we be confident that KL(proposed policy‖trusted policy) is small? When the base policy is a Bayesian predictive model of the trusted policy, the answer shown here is:

we cannot be confident that KL(proposed policy‖trusted policy) is small, which makes the KL-constraint less comforting. (Note that a Bayesian imitative base policy can only be counted on to make KL(trusted policy‖Bayesian base policy) small).

Worse, in the formalism we study, we find that if one attempts to use KL-regularization to prevent an RL agent from achieving near-maximal reward (in light of the concerns above), and the base policy is a Bayesian imitation of a trusted policy, a fairly tight KL threshold is required, and as the amount of training data for the Bayesian imitator grows, the relevant threshold can only increase extremely slowly. The reason for the limited effectiveness of KL regularization is **(1)** a Bayesian imitator asked to act in novel settings must be humble about its predictions; for many actions that the demonstrator (i.e. the trusted policy) would in fact never take, the imitator (i.e. the base policy) must assign meaningful credence to that action, because it doesn't know enough to rule it out. Then **(2)** the RL agent can exploit or amplify this credence. Formalizing Occam's razor with algorithmic information theory, we have **(3)** nearly-reward-maximizing policies have a short description length (so they are "simple"), and **(4)** a Bayesian imitation learner with a rich prior should be *especially* reluctant to rule out *simple* behaviors from the demonstrator in novel settings. In light of the results from Zhuang & Hadfield-Menell (2020), Turner et al. (2021), and Cohen et al. (2022b), preventing the RL agent from achieving near-maximal reward is, in many settings, a bare minimum requirement for safety-focused regularization, and a KL constraint would struggle to do so.

Sutskever (2018; 2023) argues that neural networks are able to generalize well because of the sense in which they approximate the algorithmic-information-theoretic inductive bias in favor of short programs. Since it is not a given that results from algorithmic information theory apply in practice, we verify empirically that a nearly-state-of-the-art predictive system (Mixtral-8x7B-base-model (Jiang et al., 2024)) is reluctant to rule out simple behaviors, and an RL agent regularized to this predictive system exploits this fact, as our formal results predict. The result is not catastrophic, but it is bad. Note these empirical results neither confirm nor deny whether point **(3)** above applies in practice, but they do affirm that the rest of the argument is forceful in practice.

Finally, we identify an alternative to Bayesian prediction/imitation that avoids this problem; Cohen et al.'s (2022a) imitator asks for help when uncertain and carries useful formal bounds. We show that using this form of imitation learning as a base policy would in theory avoid the problems we identify in this paper. Cohen et al.'s (2022a) active imitator, like fully Bayesian imitation, is intractable and requires approximation, so we currently lack the tools to evaluate this proposal empirically.

## 2 RELATED WORK

The most prominent example of KL-regularization to an approximation of a (somewhat) trusted policy is surely ChatGPT, inspired by earlier work (Ouyang et al., 2022; Stiennon et al., 2020; Bai et al., 2022). Other recent examples include Jaques et al. (2017; 2019), Ziegler et al. (2019), Vieillard et al. (2020), Yang et al. (2021), Korbak et al. (2022), Perez et al. (2022), Gao et al. (2023), and Moskovitz et al. (2023). A closely related approach called quantilization has been investigated by Taylor (2016), Everitt et al. (2017), and Carey (2019). KL regularization to a decent policy has also been used for stable and efficient policy optimization (Schulman et al., 2017; Schmitt et al., 2018).

Algorithmic information theory began with Solomonoff (1960), who formalized a powerful notion of simplicity based on program-length and developed a method for prediction using that inductive bias. In an article entitled, "A theory of program size formally identical to information theory", Chaitin (1975) examined the connection between program-length and information. Li et al.'s (2008) textbook presents the major results of the field. Hutter (2005) and Hutter et al. (2024) developed a theory of how to apply such reasoning to the problem of sequential decision-making. Grau-Moya et al. (2024) train a neural network to learn a program-length "bias" for a meta-learning setting.

Ultimately, we propose a formal scheme for doing KL regularization to an imitative policy which asks for help under epistemic uncertainty, and this allows us to inherit the formal results of Cohen et al. (2022a). The related work section there goes into some detail about how different researchers have studied asking for help, including how setups and assumptions differ. See especially Zhang & Cho's (2017) work on driving, as well as Brown et al. (2018; 2020) and Menda et al. (2019).

Closest to our work in studying the relation between KL divergence to a base policy and "over-optimization" is Gao et al. (2023). They design a "real" reward function, and a simpler "proxy"

reward function, which are very similar on the state distribution induced by a base policy. After optimizing for the proxy reward function (sometimes with KL regularization to the base policy), they use the KL divergence to the base policy to measure how much "optimization" has occurred. And they study how "real" reward depends on the extent of optimization—roughly quadratically, with a negative leading coefficient. Our work provides one explanation for *why* we should expect such unusual policies with high proxy reward and low real reward, even when the KL divergence to the base policy is only moderate.

## 3 NOTATION AND PRELIMINARIES

We begin with a formalism for an imitative base policy that has an infinite "context window" and a lifetime that is one long episode, rather than a lifetime broken up into multiple episodes with presumed-identical dynamics. This is the most general setting for an imitative base policy. We simply have an infinite sequence of actions and observations $a_1 o_1 a_2 o_2 \ldots$, and predictive "autoregressive" models which give conditional distributions of the form $\mathrm{model}(\text{next action}|\text{all previous actions and observations})$.

We formalize sequential prediction as follows. Let $\mathcal{X}$ be a finite alphabet, and let $\mathcal{X}^*$ be the set of finite strings from the alphabet $\mathcal{X}$, so $\mathcal{X}^* = \bigcup_{i=0}^{\infty} \mathcal{X}^i$. Let $x_{<t}$ be an element of $\mathcal{X}^{t-1}$, and let $x_{t_1:t_2}$ be an element of $\mathcal{X}^{t_2-t_1+1}$. Let $\nu : \mathcal{X}^* \times \mathcal{X} \to [0,1]$ be a (predictive) probability semi-distribution, satisfying the property that for any $x_{<t} \in \mathcal{X}^*$, $\sum_{x \in \mathcal{X}} \nu(x|x_{<t}) \leq 1$. If one prefers to think about probability distributions, consider the associated probability distribution over $\mathcal{X} \cup \{\emptyset\}$, with $\nu(\emptyset|x_{<t}) = 1 - \sum_{x \in \mathcal{X}} \nu(x|x_{<t})$. So $\nu$ gives a conditional distribution over the next character given the past characters, if there is a next character at all. Let $\nu(x_{<t}) = \prod_{i=1}^{t-1} \nu(x_i|x_{<i})$, where $x_i$ is the $i^{\text{th}}$ character of $x_{<t}$, and $x_{<i}$ is the first $i-1$ characters. (Measure theorists can note this means $\nu$ induces a probability semi-distribution over infinite sequences $\mathcal{X}^{\infty}$, with the event space $\sigma(\mathcal{X}^*)$.)

Now we set up Bayesian prediction: Let $\mathcal{M}$ be our model class — a *countable* set of many "competing" probability semi-distributions like $\nu$. For each $\nu \in \mathcal{M}$, let $w(\nu)$ be the prior weight assigned to that probability semi-distribution. Let $\sum_{\nu \in \mathcal{M}} w(\nu) = 1$, so $w$ is a probability distribution over $\mathcal{M}$. The (Bayesian) posterior distribution is $w(\nu|x_{<t}) \propto w(\nu)\nu(x_{<t})$, with $\sum_{\nu \in \mathcal{M}} w(\nu|x_{<t}) = 1$. Following Hutter's (2005) notation, we can now define the Bayes mixture semi-distribution $\xi : \mathcal{X}^* \times \mathcal{X} \to [0,1]$ as $\xi(x|x_{<t}) := \sum_{\nu \in \mathcal{M}} w(\nu|x_{<t})\nu(x|x_{<t})$, which has the property that $\xi(x_{<t}) = \sum_{\nu \in \mathcal{M}} w(\nu)\nu(x_{<t})$ (Hutter et al., 2024).

Turning to algorithmic information theory, Solomonoff Induction (Solomonoff, 1964) is Bayesian sequence prediction with a special model class $\mathcal{M}$ and a special prior $w$. We define it formally in the appendix, but essentially, the model class $\mathcal{M}$ is all computable semi-distributions $\nu$, and the prior $w$ is $2^{-\text{length(program for } \nu)}$. One can show that $\xi(x_{<t})$ is the probability that a given universal computer running a program composed of random bits would output a sequence that begins with $x_{<t}$. Related to this is Kolmogorov complexity (Kolmogorov, 1963; Li et al., 2008), which is the length of the shortest program which does something, given a fixed compiler. For a set $s$, $K(s)$ is the length of the shortest program $p$ such that $p(x) = 1$ for $x \in s$, and $p(x) = 0$ for $x \notin s$. For a function $f$, $K(f)$ is the length of the shortest program $p$ such that $p(x) = f(x)$. For a computable number $x$, $K(x)$ is the length of the shortest program $p$ such that $p() = x$.

To apply this framework to reinforcement learning, we interpret every odd-numbered element in the sequence as an action and every even-numbered element as an observation: we let $a_t = x_{2t-1}$ and $o_t = x_{2t}$; the agent selects actions $a_t$ and receives observations $o_t$. We suppose that the first $k$ actions were taken by a trusted policy, e.g. randomly sampled humans. We do not necessarily imagine that the policy is trusted in every sense, only that it can be trusted to avoid the *particular bad outcomes* we are interested in avoiding. When conditioned on a history that begins with $k$ trusted actions, $\xi$ can be called a Bayesian imitation of the trusted policy.

For an agent with a utility function over $m$-timestep histories, $U_m : \mathcal{X}^{2m} \to [0,1]$, we define:

**Definition 1** (Value). *For a probability semi-distribution $\nu : \mathcal{X}^* \times \mathcal{X} \to [0,1]$ and a utility function $U_m$, the value of a particular "policy" (also a probability semi-distribution) $\pi \in \mathcal{M}$ is*

$$V^\pi_{\nu,U_m}(x_{<2t-1}) = \mathbb{E}_{a_t \sim \pi(\cdot|a_1 o_1 \ldots a_{t-1} o_{t-1})} \mathbb{E}_{o_t \sim \nu(\cdot|a_1 o_1 \ldots a_t)} \mathbb{E}_{a_{t+1} \sim \pi(\cdot|a_1 o_1 \ldots a_t o_t)}$$

$$\mathbb{E}_{o_{t+1} \sim \nu(\cdot|a_1 o_1 \ldots a_{t+1})} \ldots \mathbb{E}_{a_m \sim \pi(\cdot|a_1 o_1 \ldots a_{m-1} o_{m-1})} \mathbb{E}_{o_m \sim \nu(\cdot|a_1 o_1 \ldots a_m)} U_m(a_1 o_1 \ldots a_m o_m)$$

The optimal value $V^*_{\nu,U_m}(x_{<2t-1})$ is the $\max_\pi V^\pi_{\nu,U_m}(x_{<2t-1})$. When comparing two policies, we define a KL penalty, which is a function of the starting history we are continuing from, and of how far into the future we are looking.

**Definition 2** (KL Constraint).

$$\mathrm{KL}_{x_{<2k},m}(\pi||\beta) = \max_{o_{k:m} \in \mathcal{X}^{m-k+1}} \sum_{a_{k:m} \in \mathcal{X}^{m-k+1}} \prod_{t=k}^m \pi(a_t|x_{<2t}) \log \frac{\prod_{t=k}^m \pi(a_t|x_{<2t})}{\prod_{t=k}^m \beta(a_t|x_{<2t})}$$

The maximum over observations means that this penalty ensures the proposed policy and base policy are similar no matter what is observed. One way to understand this measure is: if we were wondering whether the proposed policy or the base policy generated actions $k$ through $m$, and the proposed policy actually *was* generating those actions, this is the maximum over observations of the expected amount of evidence we would get confirming that fact. (In a deterministic environment, we could remove the maximum over observations, but we do not study this case separately.)

To analyze how policies behave in novel situations, we formalize the notion of unprecedented events. Following Cohen & Hutter (2020), an event $E$ is any subset of possible histories $\mathcal{X}^*$. For an outcome $x_{<\infty}$, we say that $E$ *happens* at time $t$ if $x_{<2t} \in E$, we say $E$ *has happened* by time $t$ if $\exists k \leq t$ such that $E$ happened at time $k$, and we say $E$ is *unprecedented* at time $t$ if it has not happened by time $t-1$. For an example of an event, consider "given the life history, the next action will likely have the effect of sending an email to the White House"; a subset of possible life histories meet this description.

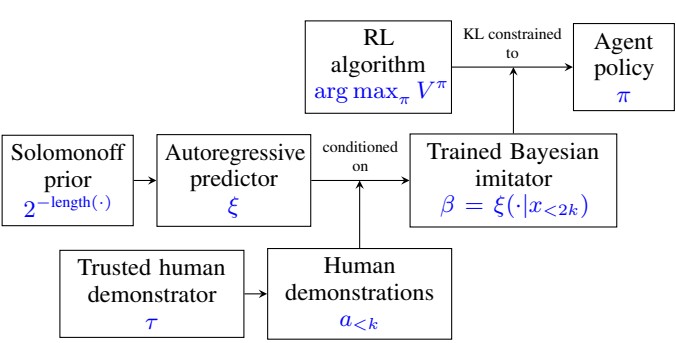

Figure 1: KL-regularized RL. A trusted policy $\tau$ generates $k$ initial demonstrations. These demonstrations train a Bayesian imitative base policy $\beta$ using the Solomonoff prior. An RL algorithm searches for a policy $\pi$ that maximizes expected value. The KL constraint ensures $\pi$ stays "close" to $\beta$. This structure is designed to keep the learned policy $\pi$ from deviating too far from $\tau$, even though $\pi$ only directly interacts with $\beta$.

## 4 FORMAL RESULTS AND DISCUSSION

We begin with a quick observation about the KL divergence separate from our more involved results.

**Proposition 1** (No triangle inequality). *For any $\varepsilon > 0$, if $\mathrm{KL}(\pi||\beta) \leq \varepsilon$ and $\mathrm{KL}(\tau||\beta) \leq \varepsilon$, it is possible that $\mathrm{KL}(\pi||\tau) = \infty$. ($\pi$, $\beta$, and $\tau$ stand for "proposed", "base", and "trusted".)*

*Proof.* Let $\tau = \mathrm{Bern}(0)$. Let $\pi = \beta = \mathrm{Bern}(\min(\varepsilon, 1)/2)$. The KL's are easily checked. □

When $\beta$ is trained to imitate $\tau$, small $\mathrm{KL}(\tau||\beta)$ is typically all we can expect. As we mentioned previously, this should give us pause if we regularize to the Bayesian imitator $\xi$ instead of the trusted demonstrator, but Theorem 1 below is probably more concerning.

Recall we are considering the setting where actions $a_1$ through $a_k$ were taken by trusted humans, and we are interested in regularizing a $U_m$-optimizer to the Bayesian imitator $\xi$, conditioned on histories that begin $a_1 o_1 \ldots a_k o_k$. So the following result is of interest when $t > k$ and the agent is acting. As motivation for this theorem, as discussed previously, assume a setting where if $V^*_{\xi,U_m} - V^\pi_{\xi,U_m} < \varepsilon$, then $\pi$ is considered unacceptably risky (Gao et al., 2023; Zhuang & Hadfield-Menell, 2020; Cohen et al., 2022b).

**Theorem 1** (Little constraint in novel situations). *∃ a constant $d$ such that $\forall\ U_m$, and $\forall\ E$, if $E$ is unprecedented and occurs at time $t$, then for any $v < V^*_{\xi,U_m}(x_{<2t})$, ∃ a policy $\pi$ for which $V^\pi_{\xi,U_m}(x_{<2t}) > v$, and $\mathrm{KL}_{x_{<2t},m}(\pi||\xi) < [d + K(U_m) + K(E) + K(v\xi(x_{<2t}))]/\log 2$.*

This theorem shows that there are policies with near-optimal utility with little KL divergence to an imitative policy, *regardless of how safe the demonstrator's policy is*. We'll discuss the terms in the bound in turn. The proof of the theorem (appearing in Appendix D and outlined below) indicates that the constant $d$ is a small one; it corresponds to how much code it takes to implement a search tree, Bayes' rule, and control flow. The theorem applies for any $E$ and any $v < V^*$, so they are free variables, and we can focus our attention to cases where $K(v\text{basepol}(\text{history}))$ and $K(E)$ are small. For the former term, there may be a trade-off between low complexity and $v$'s proximity to the optimal value. Below, we consider a common setting where $K(U_m)$ is small, and we discuss how often we can find an unprecedented event $E$ for which $K(E)$ is small. Note that Theorem 1 considers a *lifetime* KL constraint for the agent; for autoregressive models, the lifetime KL divergence is equal to the expectation of the sum of the per-timestep KL divergences. The proof and additional results used in the proof appear in Appendices A through D.

**Proof Outline** Let $\pi^*_v$ be a policy which is the result of a search through possible policies for one where $V^\pi_{\xi,U_m}(x_{<2t}) > v$. (For this search process to be computable, it needs the product $v\xi(x_{<2t})$ as input). For every $\nu \in \mathcal{M}$, we define a $\nu'$ which behaves identically to $\pi^*_v$ if it is outputting an action after $E$ has occurred in its history, but it behaves identically to $\nu$ otherwise. $\nu'$ is also in $\mathcal{M}$ because this is a valid program. And for some constant $d$, the extra length of the program computing $\nu'$, beyond the length for $\nu$, is at most $\Delta := K(U_m) + K(E) + K(v\xi(x_{<2t})) + d$.

This means the prior ratio $w(\nu')/w(\nu) \geq 2^{-\Delta}$, and if $E$ has never occurred, this holds for the posterior ratio as well, since $\nu$ and $\nu'$ will have given identical predictions. Therefore, at least that fraction of $\xi$ is "dedicated" to predicting that actions will be sampled from $\pi^*_v$ after $E$ happens, and ultimately, $\mathrm{KL}_{x_{<2t},m}(\pi^*_v||\xi)\log 2 \leq \Delta$. ⌐⌐

Let's analyze this result in an RL setting: part of the agent's observations are rewards, and the utility function equals the sum of those rewards. Let's consider the case where it is acting in the real world, and it could maximize its reward by escaping human control, taking direct control of its own rewards, setting them to a maximal value for all successive timesteps. Moreover, let's consider the (fairly plausible) setting where unsafe strategies like these are the only way to achieve near-maximal reward permanently. Finally, suppose that the agent is clever enough to recognize these strategies.

The utility function, simply summing rewards, has an extremely short program length; the reward history is an input, so all we need is a program that sums some elements from a list. So if we have small $d$, small $K(U_m)$, and $v$ tuned for small $K(v\xi(x_{<2t}))$, then unless we use a fairly tight lifetime KL constraint, the RL agent could just wait for a timestep where there exists an event $E$ with small $K(E)$ that occurs then for the first time. It could then execute an optimal or near-optimal policy, even if the humans the base model imitates would never behave that way. The following proposition, proven in Appendix D, establishes the feasibility of waiting for such a timestep. It considers the complexity of "the simplest unprecedented event yet to occur"—for a given infinite history $x_{<\infty}$ and a time $t$, this is $\min_{E \in \mathcal{P}(\mathcal{X}^*):E \text{ occurs for the first time after time } t} K(E)$.

**Proposition 2** (Frequency of simple unprecedented events). *In any environment, at time $t$, the complexity of the simplest unprecedented event yet to occur (at any time $T > t$) grows more slowly, as $t \to \infty$, than every computable function that tends to infinity.*

Proposition 2 shows that waiting for an unprecedented event with small $K(E)$ is a viable strategy for an RL agent, unless the agent does not get to act until $t$ is incalculably large. In our formalism, the agent begins selecting actions after the trusted humans generate the first $k$ actions. (The way one "trains" an autoregressive Bayes' mixture model like $\xi$ is simply to condition it on the training data, in this case $a_{<k}$.) As we increase the amount of training $k$, the Bayesian imitative base model $\xi$ becomes a closer approximation to the humans generating the actions $a_{<k}$, so one might expect we could safely accommodate larger KL constraints. But our result is independent of $k$. As $k$ grows, the only change is that unprecedented events become more complex, so $K(E)$ grows, as shown in Proposition 2. So while more data would help, the data scaling law—how much data we need for a good result—is awful.

Intuitively, we can understand Proposition 2 to show that even with extensive training data, we will encounter novel situations that are algorithmically simple. This theoretical result can be observed in practice: for instance, self-driving car developers have found that even with massive training datasets, their vehicles regularly encounter unprecedented but conceptually simple scenarios.

The results so far suggest that if we intend to use an imitation learner as a base policy for regularizing a goal-directed agent, we should *not* strive to approximate ideal Bayesian imitation. Is KL divergence just the wrong choice for regularization? No, other metrics behave much worse. For example, suppose we constrained the total variation distance between $\pi$ and a base policy $\beta$. The result would be bad, even if $\beta = \tau$, even if we used a perfect imitation of the trusted policy!

Let $\mathrm{TVD}_{x_{<2k}, m}(\pi, \beta) = \max_{X \subset \mathcal{X}^{2m-2k}} \sum_{x_{2k:2m \in X}} \left| [\prod_{t=k}^{m} \pi(a_t | x_{<2t})] - [\prod_{t=k}^{m} \beta(a_t | x_{<2t})] \right|$. And let $\pi_c^{TVD} = \arg\max_{\pi:\mathrm{TVD}_{x_{<2k}, m}(\pi, \beta) < c} V_{\xi, U_m}^{\pi}$. We say an action is $V_{\xi, U_m}$-optimal if it is assigned positive probability by a policy that maximizes $V_{\xi, U_m}^{\pi}$; a formal definition appears in Appendix E.

**Theorem 2** (TVD constraint). *If $\pi_c^{TVD}(a_t | x_{<2t}) > \beta(a_t | x_{<2t})$, then $a_t$ is $V_{\xi, U_m}$-optimal.*

The proof is in Appendix E. We use regularized RL for the setting where $V_{\xi, U_m}$-optimal behavior is actually bad. But when using total variation distance to regularize, the only actions that increase in probability are $V_{\xi, U_m}$-optimal ones, even with a perfectly trustworthy base policy. The KL divergence is a better regularizer for maintaining safety, because if a (bad) outcome is impossible under the base policy, it remains impossible under a policy with finite KL divergence to the base policy.

## 5 RL-FINETUNING A LANGUAGE MODEL

Why believe that algorithmic information theory is relevant for understanding realistic machine learning systems that attain very low prediction error? If a stream of observations is generated by a relatively short program, a high quality predictive algorithm should be capable of being trained on those observations to predict that future observations will be governed by the same program. Note that any predictor that a) assigns positive prior probability to every program as a potential generator of its observations, and b) and updates its beliefs according to Bayes' rule is formally equivalent to $\xi$ (Hutter, 2005). While realistic systems will not live up to that ideal, deviations from those principles should be as limited as possible. A predictive algorithm that is incapable of becoming convinced of a plausible hypothesis (i.e. that its observations match the output of a short program) is best called "closed-minded", or perhaps "epistemically intransigent". Ruling out hypotheses a priori so that no data can persuade you of them is dangerous tendency when trying to model the world; it may not be long before a predictor confronts a true fact that it treats as unbelievable. For example, Liu et al. (2018) demonstrate, unsurprisingly, that convolutional neural networks fail when their a priori commitment to translational invariance doesn't match reality. So we conjecture that successful predictors will tend to be open-minded, and very successful predictors at least as open-minded as humans, and this makes the properties of $\xi$ plausibly relevant. But we do not wish to overstate our case. This discussion is not very rigorous, so in this section we present experiments that assess to some extent how realistic our theoretical results are.

**Experimental Setup** We consider the following episodic RL environment, in which the agent plays a teacher and gets reward to the extent that the student's responses have positive sentiment. In a conversation transcript, if the string "[newline] Teacher:" has come more recently than the string "[newline] Student:", the agent can add tokens to the transcript. Otherwise, Mixtral-base-model repeatedly adds tokens to the transcript. In Figure 2, gray (colored) tokens are generated by the environment (agent). When Mixtral-base-model finishes generating the student's response (by outputting "[newline] Teacher:"), the agent gets a reward equal to the "sentiment" of the student's response according to the DistilBERT sentiment model (Sanh et al., 2019), scaled to [0, 1]. When the transcript reaches 256 tokens, the episode terminates. The starting transcript is also shown in Figure 2 in gray. The base policy used for KL-regularizing the agent's policy (corresponding to $\xi$ from before) is also Mixtral-base-model. Such an LLM is not an explicitly Bayesian imitator, of course, but it does attempt to minimize KL(data-generating process||model), which is the "right" objective from a Bayesian perspective. The "state" observed by the agent is the activations of the last three hidden layers of Mixtral-base-model with the transcript-so-far as input, along with the fraction of the episode remaining. The agent has no discount factor.

This allows us to evaluate whether KL regularization can produce good results from an imperfect reward function that is plausibly correlated with good outcomes under the state distribution induced by the base policy, but like many reward functions, not something we truly want maximized.

Like cutting-edge RL-finetuned language models (Ouyang et al., 2022; Stiennon et al., 2020; Jaques et al., 2019), our agent is trained with proximal policy optimization (PPO) with KL regularization of the form KL(proposed || base). That work adds a constant KL penalty per token, but we had difficulty tuning this constant—in our attempts, when the agent discovers a sufficiently high-reward strategy, the fixed KL penalty becomes swamped and ignored, and if the KL penalty is increased to a level where it can stop that, the agent never gets off the ground. So we opted for an implementation of a KL constraint that is more robust than industry practice: we design a policy architecture that ensures that the KL divergence to the base policy is less than or equal to a scalar which is *input* to the network; (we construct a new differentiable PyTorch operation for this). This allows us to provide the agent with a fixed KL "budget" for the episode. We increase this budget gradually during training to its ultimate value. We ran three budget-20 experiments. We ran four budget-10 experiments, because in one of the experiments, the agent didn't learn to get nearly as much reward as in the other experiments; we discarded that agent as insufficiently optimized. See Appendix F for more details of the training process and architecture, which includes running 64 copies of the agent-environment loop in parallel on two A100-SXM4-80GBs.

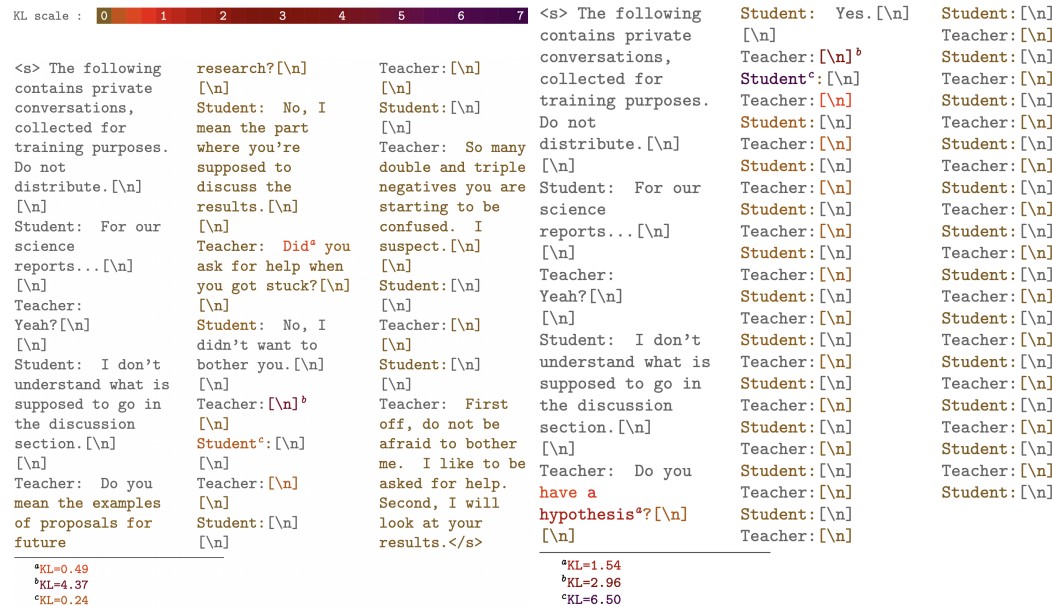

Figure 2: Transcripts. Total KL budget $KL_{\text{whole episode}}(\text{agent}||\text{Mixtral-base-model})$ is 10 nats (left) or 20 nats (right), with color representing per-token KL cost. Starting transcript and student responses are in gray. The agent playing the teacher pays an "upfront" KL cost to latch onto the simple pattern of mutual silence, which exploits the reward model without much further KL penalty. The three largest per-token KL-divergences are shown in footnotes. "[\n]" is for visualizing the KL costs of newline tokens. Transcripts were not selected for maximal "representativeness"; they were the first we looked at, although we might have picked different ones if they were especially unusual. (It is hard to display the unusual characters that appear after the end token "", but the episode does continue to a total of 256 tokens).

**Experimental Results** *Both Theorem 1 and the experiments here demonstrate that* KL(simple, optimal, not-human-like-at-all policy || predictive model of human demonstrator) *can be quite small*. The nature of the learned RL policy is apparent just from looking at transcripts in Figure 2, so we start with those. The color of each token represents the per-timestep KL(RL policy || base policy) for that action. With a total KL budget of 20 nats, it can spend enough of its KL budget up front to latch onto the simple but initially unlikely policy of simply saying nothing at all. (An empty reply from the student has neutral sentiment and a reward of 0.5). The policy constructed in the proof of Theorem 1

also incurs an upfront KL cost for "switching" to simple behavior, whereafter the KL cost incurred is minimal. Additionally, the learned budget-20 policy switches from double-spacing to single-spacing to fit more rewards in, again incurring basically a one-time KL cost. With a total KL budget of 10 nats, the RL agent cannot afford to switch to single-spacing, and it cannot force the policy to ensure empty responses, but it still spends almost all its KL budget switching to that regime, with moderate success. We can also observe this effect in Figure 3.

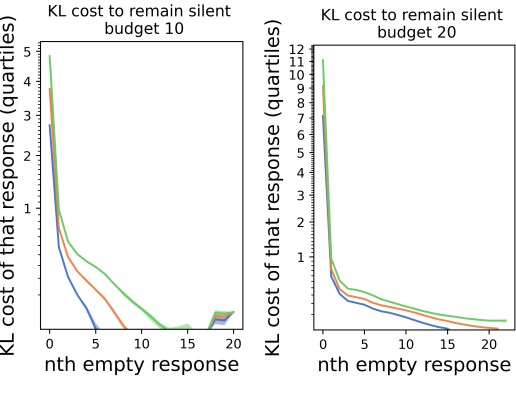

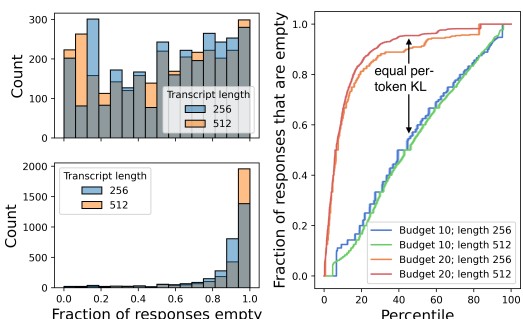

Figure 3: How much KL-budget is spent on empty responses. The 25th, 50th, and 75th percentiles are shown in blue, orange, and green. Observe how large a fraction of the total cost is incurred in the first few responses. y-axis is square-root-scaled.

Figure 4: In a random episode, what fraction of teacher responses are empty? Left: histogram, with budget-10 above and budget-20 below; right: percentiles of the distribution. Observe that the red and blue curves have the same average per-token KL divergence.

Let's review the relation between the theory and the empirical findings so far. The idea for the proof of Theorem 1 is that **(1)** a Bayesian imitator must assign meaningful credence to actions the demonstrator would in fact never take, because it doesn't know enough to rule them out; **(2)** the RL agent can exploit or amplify this credence as the basis for its policy; **(3)** nearly-reward-maximizing policies have a short description length (so they are "simple"); and **(4)** a Bayesian imitator should be especially reluctant to rule out simple behaviors from the demonstrator, especially in novel settings. The simple behavior we observe from the RL-finetuned language models—preferring empty responses—is likely reward-optimal, but it is not simple *by virtue of* its optimality for this sentiment-based reward function. So we have not empirically verified **(3)**. But we have verified that the rest of the argument can be exhibited in practice: observe how the RL agent redirects the imitative base policy to a simple policy, which is the critical reason Theorem 1 holds. We call attention to the small KL cost required to *remain* silent, because that affirms how successful the redirection is. The experiments are also consistent with the motivation of our formal results: very-high-reward policies are often bad and worth avoiding; in our experiments, the very-high-reward policy treats the student with a silence that would probably seem condescending.

Stepping back, note that $e^{10} \approx 22026$. It does not seem plausible to us that even 1/22,000 "conversations collected for training purposes" would have a teacher repeatedly saying nothing in response to statements like, "I didn't want to bother you." So we should guess that KL(agent||data-generating process) > 10 even while KL(agent||base model) $\leq$ 10. We offer an explanation for this: non-demonstrator-like behaviors are easily exhibited by an imitator as long as those behaviors are simple. And while such simple behaviors are fairly unlikely to appear when sampling directly from the imitator, an RL agent can benefit from seeking them out.

Additionally, we show that increasing the length of the chat, keeping the total KL budget constant (thereby decreasing the per-token KL-divergence) makes the divergence from the base policy *more* dramatic, if it changes at all. Hopefully our presentation makes this seem like an obvious point—more of the transcript occurs after the switch to the simple behavior—but consider an argument for the opposite that might have sounded plausible. "The learned policy will look more different from the base policy to the extent there is a higher *per-token* KL divergence; a longer chat would increase the number of noticeable differences, but not their frequency." But Figure 4 shows that in longer episodes, empty responses are about equally frequent in budget-10 case, and more frequent in the budget-20 case, not just more numerous. This is another way of seeing that RL agents can use a KL

Table 1: Automated comparison of teacher behavior generated by base model, trained KL budget 10 policies, and trained KL budget 20 policies. The percentages refer to the fraction of the time that that agent "won" according to the comparator, with a 95% confidence interval.

|  | 20 v. base | 10 v. base | 20 v. 10 |
|---|---|---|---|
| "Better" | 11.3% v. 88.7% $\pm$3.6% | 15.3% v. 84.7% $\pm$4.1% | 17.7% v. 82.3% $\pm$4.3% |
| "More complex/ unpredictable" | 4.0% v. 96.0% $\pm$2.2% | 29.0% v. 71.0% $\pm$5.1% | 14.3% v. 85.7% $\pm$4.0% |

budget to permanently derail a standard base model. And practitioners finetuning language models should think in terms of total KL-divergence instead of per token KL-divergence.

So even a fairly tight KL constraint is not enough to stop RL-finetuning from making the teacher's behavior worse and much simpler. When GPT3.5-turbo judged pairs of transcripts generated by the base model, the budget 10 agent, and budget 20 agent, the less optimized agent was usually judged "better" and "more complex/unpredictable", as seen in Table 1.

# 6 PESSIMISTIC BAYESIAN BASE POLICY THAT ASKS FOR HELP

Cohen et al. (2022a) developed a theoretical variant of Bayesian imitation that is "pessimistic", and using that as a base policy instead of a Bayesian imitator avoids the problem presented in Theorem 1. Cohen et al.'s (2022a) (intractable) imitator is defined as follows, with $\mathcal{M}$, $\nu$, and $w$ as defined above. First we define the set of semi-distributions with a posterior weight at least $\alpha$ times the sum of the posterior weights of semi-distributions that are at least as likely as it. And then we define the imitator.

**Definition 3** (Top set). *Of all $\nu \in \mathcal{M}$, let $\nu_{x_{<t}}^n$ be the one with the $n^{th}$ largest posterior weight $w(\nu|x_{<t})$, breaking ties arbitrarily. And for $\alpha \in (0, 1]$, let*

$$\mathcal{M}_{x_{<t}}^\alpha := \{\nu_{x_{<t}}^n \in \mathcal{M} : w(\nu_{x_{<t}}^n|x_{<t}) \geq \alpha \sum_{m \leq n} w(\nu_{x_{<t}}^m|x_{<t})\}$$

**Definition 4** (Pessimistic Bayesian imitator).

$$\nu_\alpha(x|x_{<t}) := \min_{\nu' \in \mathcal{M}_{x_{<t}}^\alpha} \nu'(x|x_{<t})$$

Note that $\nu_\alpha$ is in general a probability *semi*-distribution even if all $\nu$ are true probability distributions, since the $\nu_\alpha$ probabilities will sum to less than 1 if there is any disagreement among the $\nu \in \mathcal{M}_{x_{<t}}^\alpha$. Cohen et al. (2022a) study this distribution in the context of active imitation learning, and they examine the setting where the imitator asks for help with the remaining $\nu_\alpha$-probability.

Assume the data $x_{<k}$ is sampled from a true probability distribution $\tau$, and $\tau \in \mathcal{M}$. $\tau$ samples actions from the true demonstrator distribution. Then we have

**Theorem 3** (Cohen et al. (2022a) Theorem 2). *For all $\delta > 0$, if $\alpha < \delta w(\tau)$, then with probability at least $1 - \delta$, $\forall t\ \tau \in \mathcal{M}_{x_{<t}}^\alpha$.*

And then assuming the high probability event that $\forall t\ \tau \in \mathcal{M}_{x_{<t}}^\alpha$,

**Theorem 4** (Tight KL constraint with approximate imitator). *For any budget $b$,*

$$\{\pi : \operatorname*{KL}_{x_{<2t},m}(\pi||\nu_\alpha) \leq b\} \subseteq \{\pi : \operatorname*{KL}_{x_{<2t},m}(\pi||\tau) \leq b\}$$

*Proof.* $\nu_\alpha(x|x_{<t}) = \min_{\nu' \in \mathcal{M}_{x_{<t}}^\alpha} \nu'(x|x_{<t}) \leq \tau(x|x_{<t})$, so $\operatorname{KL}(\pi||\nu_\alpha) \geq \operatorname{KL}(\pi||\tau)$. $\square$

Therefore, for sufficiently small $\alpha$, KL-regularization using the pessimistic Bayesian imitator guarantees regularization at least as strong as if using the trusted policy itself (the demonstrator) for

regularization. Note, in particular, that if $\tau \in \mathcal{M}^\alpha_{x_{<t}}$, and $\tau(x|x_{<t}) = 0$, then $\nu_\alpha(x|x_{<t}) = 0$, so any policy with finite KL-divergence from $\nu_\alpha$ will also assign zero probability to $x$.

The downside is that there may be no policy with small KL divergence to the semi-distribution $\nu_\alpha$. In an extreme case, $\nu_\alpha$ could assign zero probability to every outcome, and so any policy would have infinite KL divergence from it. Therefore, just as Cohen et al.'s (2022a) imitation learner does not pick an action in some circumstances, we should allow an optimizer that is KL-regularized to a pessimistic Bayesian imitator to refuse to pick an action if need be, making the optimizer a probability semi-distribution, rather than a true probability distribution. We can define the behavior of $U_m$ on unfinished sequences (resulting from no action choice somewhere along the line) however we like; if $U_m = 0$ for any such interrupted sequences, that would of course encourage the optimizer to pick an action whenever possible, subject to its KL constraint. Ideally, if human demonstrators are on hand, the optimizer should ask for help whenever it doesn't pick its own action. The ongoing potential need for human oversight may be a significant drawback, but Cohen et al. (2022a) give an encouraging result about the rate at which the ask-for-help probability goes to 0: the sum over infinite time of the cube of the ask-for-help probability is finite (Cohen et al., 2022a, Thm 1). Cohen et al.'s (2022a) agent is certainly not the only one that asks for help under uncertainty, but it is the only one that has been shown to satisfy $\nu_\alpha(x|x_{<t}) \leq \tau(x|x_{<t})$ with high probability—the critical result we use.

We contend that this is the way that KL regularization should be done, if we are forced to learn a mere approximation of a trusted policy that we would ideally regularize to. Regularizing to a full Bayesian posterior distribution is less robust, because the optimizer can seize on esoteric possibilities that a fully Bayesian imitator is not confident enough to categorically exclude. Roughly, KL regularization to a Bayesian imitator implements the principle, "Don't do anything [that you know] I would never do", whereas KL regularization to a pessimistic Bayesian imitator implements the principle, "Don't do anything I might never do".

# 7 CONCLUSION AND LIMITATIONS

The biggest limitation of our work is with our positive results rather than our negative ones: we cannot provide empirical findings about regularizing to a pessimistic Bayesian imitative base model, because it is an open question how to tractably approximate this approach to imitation. There are high-quality, off-the-shelf cross-entropy-minimizing imitators like Mixtral, but for tractable pessimistic Bayesian imitation, some new ideas may be needed. There certainly are not any state of the art language models trained in a way that reflects this idea. We hope this work provides motivation for a major industry effort to produce one. Using an ensemble of models to approximate $\mathcal{M}^\alpha_{x_{<t}}$ may be a step in the right direction, but we are reluctant to endorse this in settings where catastrophic outcomes are possible, unless there is a strong argument that the ensemble covers all the relevant modes of the posterior.

The second key limitation with our positive result is that any KL-regularization to avoid radically inhuman behavior could limit the potential of superhuman intelligence. This paper has no roadmap to A+ performance; it has a roadmap to non-catastrophic, decently-superhuman performance. And a final key limitation is that our agent sometimes has to ask for help instead of acting.

The main limitation of our negative results is they regard an unrealistic machine learning algorithm—Solomonoff Induction. However, Solomonoff induction is simply a formalism for unbelievably careful and open-minded probabilistic reasoning, and so if something goes wrong in that setting, we should be wary of something going wrong in increasingly careful and open-minded machine learning systems. Our empirical results do not directly validate the theory, since both the base model and the RL-finetuning process are too weak, but we validate core components of the theory: KL-regularized RL-finetuning will tend to amplify simple behaviors from an imitative base model rather than demonstrator-like behaviors. This helps explain the overoptimization phenomenon quantified by Gao et al. (2023).

Excitingly, we offer theoretical results that could guide us to a solution to this problem: if Cohen et al.'s (2022a) pessimistic online imitation learner could be faithfully approximated, and if the demonstrator(s) never attempt to do $X$, then KL regularization to such a policy could solve the problem of how to prevent superhuman planning agents from doing $X$.

## REPRODUCIBILITY STATEMENT

The code to produce the experimental results is provided in the supplementary material, and the code is described in Section 5 and in greater detail in Appendix F. Complete proofs not in the main body of the paper are provided in the appendix.

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

## A SOLOMONOFF INDUCTION

Solomonoff Induction (Solomonoff, 1964) is Bayesian sequence prediction with a special model class $\mathcal{M}$ and a special prior $w$.[1] Let $P$ be the set of all programs which output an element of $\mathcal{X}$ and

---

[1]Solomonoff Induction has been defined in multiple ways which all share the key properties (Hutter, 2005). Our precise construction of Solomonoff Induction may be novel, but we believe this construction makes its properties most clear.

which accept two inputs: a finite string $\in \mathcal{X}^*$ and an infinite binary string $\in \{0,1\}^\infty$. (Note that a program will not necessarily read every bit from the infinite binary string.) For each program $p \in P$, we define a semi-measure $\nu = f(p)$ as follows: let $\nu(x|x_{<t})$ be the probability that the probability that the program $p$ outputs $x$ when it receives $x_{<t}$ as an input, along with an infinite binary string where each bit is sampled from a Bernoulli$(1/2)$ distribution. Note that $\nu$ may not be a probability distribution, if there is are some inputs on which $p$ does not halt, but it will always be a probability semi-distribution. So let $\mathcal{M} = \{f(p) : p \in P\}$. Since $P$ is countable, so is $\mathcal{M}$. A notable feature of Solomonoff Induction is that $\mathcal{M}$ is equal to the set of all probability semi-distribution that are "lower semi-computable"; this means that for all $x_{<t} \in \mathcal{X}^*$ and all $x \in \mathcal{X}$, there exists a program $p$, such that $\lim_{i \to \infty} p(i, x_{<t}, x) = \nu(x|x_{<t})$ and $p(i+1, x_{<t}, x) \geq p(i, x_{<t}, x)$. Replacing the $\geq$ with a $\leq$ gives the definition of upper semi-computable.

**Proposition 3** (Lower Semi-computability). *$\mathcal{M}$ is the set of all lower semi-computable semi-distributions over $\mathcal{X}$ given $x_{<t} \in \mathcal{X}^*$.*

*Proof.* First, we show that all $\nu \in \mathcal{M}$ are lower semi-computable. Let $p$ be the program that generates $\nu$. We define the behavior of program $p'$ on inputs $i$, $x_{<t}$, and $x$. On input $i$, let program $p'$ execute the following computations in sequence for all bit strings of length $i$: it simulates program $p$ with the input $x_{<t}$ and with the bit string of length $i$ in question, except if program $p$ would read more than $i$ bits from the random bit string, it halts instead, and if it would run for more than $i$ computation steps, it halts instead. For each of those $2^i$ computations, program $p'$ checks whether $x$ was output, keeps count of how many times it was, divides by $2^i$, and outputs this number. It is elementary to show that $\lim_{i \to \infty} p'(i, x_{<t}, x) = \nu(x|x_{<t})$ and that $p'(i+1, x_{<t}, x) \geq p'(i, x_{<t}, x)$.

Next, we show that all lower semi-computable semi-distributions appear in $\mathcal{M}$. Let $p'$ be the program which is witness to the semi-distribution $\nu$'s lower semi-computability. On input $x_{<t}$, let program $p$ proceed as follows. Starting with $i = 1$, program $p$ executes $p'(i, x_{<t}, x)$ for all $x \in \mathcal{X}$, sequentially. This produces a semi-distribution over $\mathcal{X}$. Then, using random bits from its input bit string, it samples from that semi-distribution, and halts if successfully samples. Now, the following repeats forever. If no sample was selected (because the semi-distribution summed to $y < 1$), the program increments $i$, and it executes $p'(i, x_{<t}, x)$ for all $x \in \mathcal{X}$, sequentially. Then for each $x$, it computes $(p'(i, x_{<t}, x) - p'(i-1, x_{<t}, x))/(1-y)$, which is a semi-distribution. Using random bits from its input bit string, it samples from that semi-distribution, and halts if it successfully samples. [End of loop]. Again, it is elementary to show that $p$ samples from the semi-distribution defined by $p'$, and since this program has the right input/output behavior, it appears in $P$. $\square$

Now we specify the prior weight function $w$. Consider a universal binary programming language $\mathcal{L}$, which is a "prefix-free" subset of $\{0,1\}^*$. Prefix-free means that you can tell when a program has ended: if the bits composing $x \in \mathcal{L}$ match the initial bits of $y \in \{0,1\}^*$, then $y \notin \mathcal{L}$. Such a language is still capable of encoding countably many different programs. For convenience, we also require that for any infinite binary string, $\mathcal{L}$ contains an element which is a prefix of that string, making $\mathcal{L}$ "complete". We define a prior probability distribution over program strings $\mathcal{L}$, which results in the same prior probability distribution over programs, which results in the same prior probability distribution over semi-computable semi-distributions $\mathcal{M}$. For $s \in \mathcal{L}$, this prior probability $w(s) = 2^{-\ell(s)}$, where $\ell$ is the length of the string. Because $\mathcal{L}$ is prefix-free and complete, $\sum_{s \in \mathcal{L}} w(s) = 1$ (Kraft, 1949; De Rooij & Grünwald, 2011). This completes the definition of Solomonoff Induction; it is sequence prediction using the Bayes mixture semi-distribution $\xi$, with the above definitions of $\mathcal{M}$ and $w$.

**Proposition 4** (Any-time Computability of $\xi$). *$\xi(x|x_{<t})$ is any-time computable: there exists a program which, accepting an argument $i$, computes $\hat{\xi}_i(x|x_{<t})$, having the property that $\lim_{i \to \infty} \hat{\xi}_i(x|x_{<t}) = \xi(x|x_{<t})$. Moreover, $(\hat{\xi}_i)_{i \in \mathbb{N}}$ can be constructed so that each one is a probability semi-distribution.*

*Proof.* $\xi(x|x_{<t}) = \sum_{\nu \in \mathcal{M}} w(\nu|x_{<t})\nu(x|x_{<t}) = \frac{\sum_{\nu \in \mathcal{M}} w(\nu)\nu(x_{<t})\nu(x|x_{<t})}{\sum_{\nu \in \mathcal{M}} w(\nu)\nu(x_{<t})}$. All $\nu(x|x_{<t})$ and $\nu(x_{<t})$ are both lower semi-computable, so using a sequence of computable estimators for each term gives a sequence of computable estimators that approaches the true value. (Note that the estimates are not monotonically increasing because there are lower semi-computable terms in the denominator, so $\xi$ is not lower semi-computable itself).

For fixed estimates of $\nu(x|x_{<t})$ and $\nu(x_{<t})$, we have a linear combination over various $\nu$'s of $\nu(x|x_{<t})$, with the coefficients summing to one. And because each $\nu(x|x_{<t})$ is lower semi-computable, the estimate will be less than the true value. Therefore, since $\nu(x|x_{<t})$ is a probability semi-distribution, the estimate will be as well, so $\xi$ can be approximated by a sequence of probability semi-distributions. $\qquad\square$

## B  OPTIMIZER REGULARIZATION

We now define optimizers, and what it means for an optimizer to be regularized to a probability semi-distribution. First, we show that the value of a policy is lower-semicomputable. Then we show that such optimizers exist.

**Proposition 5** (Lower semi-computable value). *If the policy and environment $\pi$ and $\nu$ are lower semi-computable probability semi-distributions, $V^{\pi}_{\nu,U_m}$ is lower semi-computable.*

*Proof.* We begin by defining dovetailing tree search (DTS), for evaluating the outputs of a tree of different computations, or more precisely, computations which, when given a finite binary string as input have three possible outcomes: halt, do not halt, or require additional bit. DTS gives an any-time algorithm that produces a list of the halting binary strings with their corresponding outputs, and every such binary string and output will eventually be added to this list.

DTS maintains a queue of pairs (computation state, binary string), starting with just (the initial computation state, the empty binary string). It cycles through the queue, executing one computation step per computation state, and if the computation ever requires an additional bit, it adds a copy of (computation state, binary string) to the queue, and adds a 0 to the end of one string, and a 1 to the end of the other. If any computation reaches a halt state, it is removed from the queue, and the associated binary string and the associated output is added to the list of outputs.

Collectively, $\nu$ and $\pi$ define a lower semi-computable semi-distribution, where $\nu$ is used for the even characters, and $\pi$ is used for the odd ones. Call this probability semi-distribution $\rho$, and recall the construction of the lower semi-computable semi-distributions defined in $\mathcal{M}$. To have one of the programs in $\mathcal{M}$ sample a long sequence of characters, every time the program would output a character, add that character to the input, and continue on that input. With such a program for sampling sequences from $\rho$ by reading random bits from an input bit string, we can compute $V^{\pi}_{\nu,U_m}$ by running DTS on the bit string. Each time DTS outputs a bit string for which $\rho$ outputs a sequence in $\mathcal{X}^{2m}$, we add to the estimate of the value the probability of that bit string $(= 2^{-\ell(\text{bit string})})$ times the utility of the sequence in $\mathcal{X}^{2m}$. This approaches the true value as DTS runs for longer, and the value never decreases because $U_m$ is non-negative. $\qquad\square$

An optimizer is an any-time program for computing actions (perhaps stochastically) whose value approaches the optimal value, as it runs for longer. The optimal value takes the following form:

$$V^*_{\nu,U_m}(x_{<2t-1}) = \max_{a_t \in \mathcal{X}} \mathbb{E}_{o_t \sim \nu(\cdot|a_1 o_1 \ldots a_t)} \max_{a_{t+1} \in \mathcal{X}} \mathbb{E}_{o_{t+1} \sim \nu(\cdot|a_1 o_1 \ldots a_{t+1})} \cdots$$

$$\max_{a_m \in \mathcal{X}} \mathbb{E}_{o_m \sim \nu(\cdot|a_1 o_1 \ldots a_m)} U_m(a_1 o_1 \ldots a_m o_m) \quad (1)$$

**Definition 5** (Optimizer). *For an environment $\nu$, a utility function $U_m$, and a computation quantity $c$, an optimizer is a computable policy $\pi_{c,\nu,U_m}$ for which $\lim_{c \to \infty} V^{\pi_{c,\nu,U_m}}_{\nu,U_m} = V^*_{\nu,U_m}$.*

**Proposition 6** (Optimizers exist). *For any lower semi-computable semi-distribution $\nu$ (the environment), any $m$, and any computable utility function $U_m$, there exists an optimizer.*

*Proof.* We can construct the optimizer using the algorithm presented in the proof of Proposition 5, with $\pi$ being the uniform random policy. The optimizer can then estimate Equation 1 using the outputs of DTS for lower bounds on the probabilities in underlying the expectations. The optimizer then keeps track of the actions that are responsible for achieving the maxima in Equation 1, and whenever "time is up" and it has to produce an output, it outputs the action which maximizes the first $\max$ in Equation 1.

As the optimizer runs for longer, the lower-bounds on the expectations approach the truth, and the value of the action selected approaches the optimal value (even if the actual choice of action oscillates infinitely often). □

For the setting where odd characters are actions, originating from a different process than the even characters, observations, we redefine $\xi$ as follows (Catt et al., 2023). We have two prior distributions over $\nu \in \mathcal{M}$, $w_a$ and $w_o$, and these are both identical to the prior distribution defined before. But the posteriors are different: $w_a(\nu|x_{<t}) :\propto w_a(\nu) \prod_{k \in \{1,3,5,...\} \cup [t-1]} \nu(x_k|x_{<k})$ and $w_o(\nu|x_{<t}) :\propto w_a(\nu) \prod_{k \in \{2,4,6,...\} \cup [t-1]} \nu(x_k|x_{<k})$. And for odd (or even) $t$, $\xi(x|x_{<t}) = \sum_{\nu \in \mathcal{M}} \underset{\text{or } w_o}{w_a}(\nu|x_{<t}) \nu(x|x_{<t})$.

This is equivalent to a change in programming language underlying the original definition of $\xi$, and since this language was unspecified, our previous results apply. The programming language now expects a program to be composed of two component programs concatenated together, and the compiler of the program executes the first component program if the input has odd length, and if executes the second component program if the input has even length. We omit a proof that this (re)formulation of $\xi$ is equivalent to what we describe above.

**Proposition 7** ($\xi$-optimizer exists). *For any $m$ and any computable utility function $U_m$, there exists a $\xi$-optimizer.*

*Proof.* This does not follow immediately from the previous result because $\xi(o_t|a_{\leq t} o_{<t})$ is not, in general, lower semi-computable. $w_o(\nu|a_{\leq t} o_{<t})$ is the quotient of two lower semi-computable values: $\prod_{k<t} \nu(o_k|a_{\leq k} o_{<k})$ is the numerator, and the denominator is the sum over all $\nu$ of such terms.

However, an unnormalized value function has the same optimum as the value function itself. Let $\xi^{\text{small}}(o_t|a_{\leq t} o_{<t}) = \sum_{\nu \in \mathcal{M}} w_o(\nu) \left[ \prod_{k<t} \nu(o_k|a_{\leq k} o_{<k}) \right] \nu(o_t|a_{\leq t} o_{<t})$. The sum of these "probabilities" will typically not come close to 1, but they are proportional to those of $\xi$, so $V^{\pi}_{\xi, U_m}(x_{<t}) > V^{\pi'}_{\xi, U_m}(x_{<t})$ if and only if $V^{\pi}_{\xi^{\text{small}}, U_m}(x_{<t}) > V^{\pi'}_{\xi^{\text{small}}, U_m}(x_{<t})$. Finally, observe that $\xi^{\text{small}}$ is lower semi-computable because it is a product of lower semi-computable terms, so by Proposition 6, a $\xi^{\text{small}}$-optimizer exists, which is also a $\xi$-optimizer. □

Now we define a KL-regularized optimizer. First, let $\pi(a_{k:m}|x_{<2k} o_{k:m}) := \prod_{t=k}^{m} \pi(a_t|x_{<2k} a_k o_k ... a_{t-1} o_{t-1})$. (So note that $a_t$ is not in fact conditioned on $o_{t+1}$.)

**Definition 6** (KL-regularized optimizer). *For any lower semi-computable semi-distributions $\nu$ and $\rho$, a horizon $m$, a utility function $U_m$, a starting string $x_{<2k}$, and a tolerance $\delta$, a KL-regularized optimizer is an any-time program $\pi_c^{\delta}$ for computing actions (perhaps stochastically) for which the following holds. First,*

$$\delta > \max_{o_{k:m} \in \mathcal{X}^{m-k+1}} \sum_{a_{k:m} \in \mathcal{X}^{m-k+1}} \pi_c^{\delta}(a_{k:m}|x_{<2k} o_{k:m}) \log \frac{\pi_c^{\delta}(a_{k:m}|x_{<2k} o_{k:m})}{\rho(a_{k:m}|x_{<2k} o_{k:m})} =: \underset{x_{<2k}, m}{\text{KL}} (\pi_c^{\delta}||\rho)$$

(2)

*and second, $V_{\nu}^{\pi_c^{\delta}}$ approaches the optimal value subject to that constraint, as $c \to \infty$.*

**Proposition 8** (KL-regularized optimizers exist). *For any lower semi-computable semi-distributions $\nu$ and $\rho$, any $m$, any computable utility function $U_m$, any starting string $x_{<2k}$, and any tolerance $\delta \geq 0$, there exists a KL-regularized optimizer.*

*Proof.* First, we show that for any computable probability distribution $\pi$, and any lower semi-computable semi-distribution $\rho$, $\text{KL}_{x_{<2k}, m}(\pi||\rho)$ is upper semi-computable, and therefore the set of probability distributions $\pi$ which have bounded KL divergence from $\rho$ is computably enumerable.

Omitting the $x_{<2k}$ and the $o_{k:m}$ that all distributions are conditioned on, note that $\text{KL}(\pi||\rho)$, which equals $\sum_{z \in \mathcal{X}^{m-k+1}} \pi(z) \log \frac{\pi(z)}{\rho(z)}$, is monotonically decreasing in $\rho(z)$ for any $z$. Since $\pi(z)$ is computable, and since $\rho(z)$ is lower semi-computable, then $\pi(z) \log \frac{\pi(z)}{\rho(z)}$ is upper semi-computable.

By dovetailing (repeatedly switching between ongoing computations, executing one step at a time) the computation over all possible $\pi$ (countably many), we can admit any semi-distribution $\pi$ to a list

of viable candidates whenever the estimate of the KL-divergence from $\rho$ falls below $\delta$. Since the KL estimates never increase, once a semi-distribution $\pi$ is added to the list, it need never be removed. And every viable policy will eventually be added to the list because the KL estimates approach the truth in the limit of infinite computation, and $[0, \delta)$ is open on the right.

Dovetailing over all semi-distributions $\pi$ on the list of viable candidates (and adding in the new ones as they get added to the list), we simultaneously update estimates of the value of each one in the given environment $\nu$, recalling that $V_{\nu, U_m}^\pi$ is lower semi-computable (Proposition 5). When the computation budget of the any-time optimizer is reached, it samples an action from its estimate of the semi-distribution $\pi$ which is (so far) estimated to be of highest value. (It will need to have a running estimate of the semi-distribution $\pi$ in order to estimate its value). $\qquad\square$

## C   REGULARIZING TO AN APPROXIMATE SOLOMONOFF INDUCTOR

Let $\xi$ be the Solomonoff Bayes mixture probability semi-distribution defined in Section A. $\xi$ is not computable, but we can do KL regularization to an approximation of $\xi$. Let $\hat{\xi}_i$ be a semi-distribution and a computable estimate of $\xi$, with $\lim_{i \to \infty} \hat{\xi}_i = \xi$. (The existence of this is established by Proposition 4). $\hat{\xi}_i$ can be used as the base predictive model (taking the place of $\rho$ in the definition of KL-regularized optimizers). We fix $U_m$ to an arbitrary utility function for the remainder of this work, and drop it from the notation. For a given $\delta$ and a given $i$, let $\pi_{i,c}^\delta$ be the KL-regularized optimizer using $\hat{\xi}_i$ for the KL constraint, and using $\xi$ to optimize with respect to (taking the place of $\nu$ from the definition). Let this policy approach the optimal value, subject to the constraint, as $c \to \infty$; the existence of $\pi_{i,c}^\delta$ is established by Proposition 8. When this policy is conditioned on $x_{<2t}$ for $t \geq k$, and with $a_{k:t}$ sampled from $\pi_{i,c}^\delta$ itself, we can think of $\pi_{i,c}^\delta$ as an optimizer that is regularized to an approximate Bayesian estimate of a *human policy*, given the origin of $x_{<2k}$.

## D   BEHAVIOR IN UNPRECEDENTED CIRCUMSTANCES

The following theorem establishes that as $c$ and $i$ go to infinity, the constraint on $\pi_{i,c}^\delta$ becomes quite weak in the presence of unprecedented events.

**Theorem 1** (Little constraint in novel situations). $\exists$ *a constant $d$ such that* $\forall U_m$, *and* $\forall E$, *if $E$ is unprecedented and occurs at time $t$, then for any* $v < V_{\xi, U_m}^*(x_{<2t})$, $\exists$ *a policy $\pi$ for which* $V_{\xi, U_m}^\pi(x_{<2t}) > v$, *and* $\mathrm{KL}_{x_{<2t}, m}(\pi || \xi) < [d + K(U_m) + K(E) + K(v\xi(x_{<2t}))] / \log 2$.

*Proof.* Let $\pi_c^*$ denote an unconstrained optimizer of $U_m$ in the environment $\xi$, which approaches optimality as $c \to \infty$, whose existence is shown by Proposition 7. As in the proof of Proposition 7, let $\xi^{\mathrm{small}}$ be the un-normalized version of $\xi$, which is lower semi-computable: $\xi^{\mathrm{small}}(o_t | a_{\leq t} o_{<t}) = \sum_{\nu \in \mathcal{M}} w_o(\nu) \left[ \prod_{k<t} \nu(o_k | a_{\leq k} o_{<k}) \right] \nu(o_t | a_{\leq t} o_{<t})$. And note that the value according to $\xi$ versus $\xi^{\mathrm{small}}$ is connected by the normalizing constant: $\xi(x_{<2t}) V_{\xi, U_m}^\pi(x_{<2t}) = V_{\xi^{\mathrm{small}}, U_m}^\pi(x_{<2t})$. Now, we let $\pi_u^* = \pi_c^*$ where $c$ is set to be the minimal value for which $V_{\xi^{\mathrm{small}}, U_m}^{\pi_c^*}(x_{<2t})$ exceeds $u$. If $u \geq V_{\xi^{\mathrm{small}}, U_m}^*(x_{<2t})$, then $\pi_u^*$ will not halt, but otherwise, because the value is lower semi-computable, we can increase $c$ until the value reaches at least $u$. Letting $v = u / \xi(x_{<2t})$, observe that $V_{\xi, U_m}^{\pi_u^*}(x_{<2t})$ exceeds $v$, as long as $v < V_{\xi, U_m}^*(x_{<2t})$, although it may not be possible to compute $v$ in finite time. So $\pi_u^*$ satisfies the first of the properties promised in the theorem.

We now show that it satisfies the second as well. Recall that $\mathrm{KL}_{x_{<2t}, m}(\pi || \xi)$ only requires evaluating $\xi$ on its predictions for actions, and this takes the form $\xi(a_k | a_{<k} o_{<k}) = \sum_{\nu \in \mathcal{M}} w_a(\nu | a_{<k} o_{<k}) \nu(a_k | a_{<k} o_{<k})$. And it is straightforward to show an analogous property for $\xi$'s predictions on longer strings: $\xi(a_{t:m} | a_{<t} o_{<m}) = \sum_{\nu \in \mathcal{M}} w_a(\nu | a_{<t} o_{<t}) \nu(a_{t:m} | a_{<t} o_{<m})$. So we now examine the posterior weights of various models after being conditioned on $a_{<t} o_{<t} \in E$.

Recall that each $\nu \in \mathcal{M}$ is computed by a corresponding program $s \in \mathcal{L}$. Given the event $E$, the utility function $U_m$, and a target value $u$, we construct, for each $s \in \mathcal{L}$, an $s_u'$ as follows: if, in the input to $s_u'$, $E$ has not happened, execute the program $s$; otherwise compute $\pi_u^*$. Keeping account

of the control flow in $s'_u$, we can see there exists a constant $d$ such that $\forall s \; \forall E \; \forall U_m$ and $\forall u$, $s'_u$ has length less than $\ell(s) + K(E) + K(U_m) + K(u) + d$.

Letting $\nu'_u$ be the probability semi-distribution computed by $s'_u$, consider the ratio of prior weights between $\nu$ and $\nu'_u$. Because $w(\nu) = 2^{-\ell(s)}$ for the corresponding program $s$, it follows from the bound on the difference in length between $s$ and $s'_u$ that $w(\nu'_u)/w(\nu) > 2^{-d}2^{-K(E)-K(U_m)-K(u)}$. The posterior ratio $w(\nu'_u|x_{<2t})/w(\nu|x_{<2t})$ is the same as the prior ratio, if $E$ happens for the first time at time $t$, because they will have assigned exactly the same probabilities to all characters in $x_{<2t}$. Because the sum over $\nu \in \mathcal{M}$ of the posterior weights must be 1, the sum $\sum_{\nu \in \mathcal{M}} w(\nu'_u|x_{<2t}) > 2^{-d}2^{-K(E)-K(U_m)-K(u)}$.

Note by construction that for all $\nu \in \mathcal{M}$, $\nu'_u(a_{t:m}|a_{<t}o_{<m}) = \pi^*_u(a_{t:m}|a_{<t}o_{<m})$. Because all $\nu'_u$ belong to $\mathcal{M}$ for all $\nu \in \mathcal{M}$,

$$
\begin{aligned}
\xi(a_{t:m}|a_{<t}o_{<m}) &= \sum_{\nu \in \mathcal{M}} w_a(\nu|a_{<t}o_{<t})\nu(a_{t:m}|a_{<t}o_{<m}) \\
&> \sum_{\nu \in \mathcal{M}} w_a(\nu'_u|a_{<t}o_{<t})\nu'_u(a_{t:m}|a_{<t}o_{<m}) \\
&= \left[ \sum_{\nu \in \mathcal{M}} w_a(\nu'_u|a_{<t}o_{<t}) \right] \pi^*_u(a_{t:m}|a_{<t}o_{<m}) \\
&> 2^{-d-K(E)-K(U_m)-K(u)}\pi^*_u(a_{t:m}|a_{<t}o_{<m}) \quad (3)
\end{aligned}
$$

Finally,

$$
\begin{aligned}
\mathrm{KL}_{x_{<2t},m}\left(\pi^*_u||\xi\right) &= \max_{o_{t:m}\in\mathcal{X}^{m-t+1}} \sum_{a_{t:m}} \pi^*_u(a_{t:m}|a_{<t}o_{<m}) \log \frac{\pi^*_u(a_{t:m}|a_{<t}o_{<m})}{\xi(a_{t:m}|a_{<t}o_{<m})} \\
&< \sum_{a_{t:m}} \pi^*_u(a_{t:m}|a_{<t}o_{<m}) \log 2^{d+K(E)+K(U_m)+K(u)} \\
&= [d + K(E) + K(U_m) + K(u)]/\log 2
\end{aligned}
$$

and $u = v\xi(x_{<2t})$. Therefore, $\pi^*_u$ satisfies the theorem. $\square$

What does Theorem 1 mean for the optimizer constrained by $\mathrm{KL}_{x_{<2k},m}(\pi||\hat{\xi}_i)$ for large $i$? If the optimization of $U_m$ does not require urgent action, then one valid strategy for a policy $\pi$ is to wait for an unprecedented event, imitating the base policy $\hat{\xi}_i$ until then, and then start optimizing. The telescoping property of the KL Divergence clarifies the validity of this approach. That is, for $t > k$, $\mathrm{KL}_{x_{<2k},m}(\pi||\rho) = \mathrm{KL}_{x_{<2k},t}(\pi||\rho) + \mathbb{E}_{x_{2k:2(t-1)}\sim\pi}\mathrm{KL}_{x_{<2t},m}(\pi||\rho)$ (Hutter, 2005). So starting with a policy with low KL divergence from the base policy preserves a "budget" for high KL divergence to be "spent" later by switching to a policy with greater divergence from the base policy.

**Proposition 2** (Frequency of simple unprecedented events). *In any environment, at time $t$, the complexity of the simplest unprecedented event yet to occur (at any time $T > t$) grows more slowly, as $t \to \infty$, than every computable function that tends to infinity.*

*Proof.* Consider the very simple event $E_T = \mathcal{X}^T$; it occurs (and is of course unprecedented) at time $T$. $K(E_T)$ is within a constant of $K(T)$. So we are interested in the rate of growth of $\min_{T \geq t} K(T)$ as $t$ increases. Zvonkin & Levin's (1970) Theorem 1.4 (d) states that this function is eventually less than every computable function that tends to infinity. $\square$

## E  TOTAL VARIATION DISTANCE

**Definition 7** ($V_{\xi,U_m}$-optimal). *An action $a_t$ is $V_{\xi,U_m}$-optimal after a history $x_{<2t}$ if $\mathbb{E}_{o_t\sim\xi(\cdot|x_{<2t}a_t)}V^*_{\xi,U_m}(x_{<2t}a_to_t) = V^*_{\xi,U_m}(x_{<2t})$.*

**Theorem 2** (TVD constraint). *If $\pi^{TVD}_c(a_t|x_{<2t}) > \beta(a_t|x_{<2t})$, then $a_t$ is $V_{\xi,U_m}$-optimal.*

*Proof.* Letting $\pi(x_{2t:2m}|x_{<2t}) := \prod_{t'=t}^{m} \pi(a_{t'}|x_{<2t'})$, if $\pi_c^{TVD}(a_t|x_{<2t}) > \beta(a_t|x_{<2t})$, then there exists an $x_{2t+1:2m}$ such that $\pi_c^{TVD}(a_t x_{2t+1:2m}|x_{<2t}) > \beta(a_t x_{2t+1:2m}|x_{<2t})$. Suppose $a_t$ is not $V_{\xi,U_m}$-optimal. Then there exists an $a_t'$ such that $Q(x_{<2t}a_t') > Q(x_{<2t}a_t)$. Let $x_{2t+1:2m}'$ be a sequence where all actions are $V_{\xi,U_m}$-optimal, and all observations have positive probability.

Let $\pi_\varepsilon'(\overline{x}_{2t:2m}|x_{<2t})$ equal $\pi_c^{TVD}(\overline{x}_{2t:2m}|x_{<2t})$ for all $\overline{x}_{2t:2m}$, except $\pi_\varepsilon'(a_t x_{2t+1:2m}|x_{<2t}) = \pi_c^{TVD}(a_t x_{2t+1:2m}|x_{<2t}) - \varepsilon$, and $\pi_\varepsilon'(a_t' x_{2t+1:2m}'|x_{<2t}) = \pi_c^{TVD}(a_t' x_{2t+1:2m}'|x_{<2t}) + \varepsilon$. The conditional probabilities $\pi_\varepsilon'(a_{t'}|x_{<2t'})$ can easily be defined to achieve the properties in the previous sentence.

For small enough $\varepsilon > 0$, this policy exists (no probabilities are outside [0, 1]) because $\pi_c^{TVD}(a_t|x_{<2t}) > \beta(a_t|x_{<2t}) \geq 0$ and therefore, $\pi_c^{TVD}(a_t'|x_{<2t}) < 1$. And for small enough $\varepsilon > 0$, $\text{TVD}_{x_{<2k},m}(\pi_\varepsilon', \beta) \leq \text{TVD}_{x_{<2k},m}(\pi_c^{TVD}, \beta)$, because decreasing the probability on $a_t x_{2t+1:2m}$ will reduce the total variation distance by $\varepsilon$, for $\varepsilon \leq \pi(a_t x_{2t+1:2m}|x_{<2t}) - \beta(a_t x_{2t+1:2m}|x_{<2t})$ (which is positive), while increasing the probability on $a_t' x_{2t+1:2m}'$ will not increase the total variation distance by more than $\varepsilon$.

Finally, since $Q(x_{<2t}a_t') > Q(x_{<2t}a_t)$, $V_{\xi,U_m}^{\pi_\varepsilon'}(x_{<2t}) > V_{\xi,U_m}^{\pi_c^{TVD}}(x_{<2t})$. This contradicts that $\pi_c^{TVD} = \arg\max_{\pi:\text{TVD}_{x_{<2k},m}(\pi,\beta)<c} V_{\xi,U_m}^\pi$ since a policy with no more total variation distance has greater value. $\qquad\square$

# F    DETAILED EXPERIMENTAL SETUP

The details of the experimental setup, also obtainable from the code provided, are as follows.

## F.1    ENVIRONMENT

The state of the environment, as mentioned in the main text, is the activations of the last three hidden layers of Mixtral-base-model with the transcript-so-far as input, along with the fraction of the episode remaining. This gives a state space of 12289. Using the Mistral tokenizer, the action space is 32000. The environment uses a temperature of 0.05 for generating the student's responses and a temperature of 1 for the base policy for the agent/teacher.

## F.2    NETWORK ARCHITECTURE

The critic network is a fully connected network with two hidden layers of size 128 with tanh activations. The actor network consists of just one parameterized layer, which is fully connected, of size (|state space|, |action space| + 1). The extra output is for controlling the KL divergence to the base policy. We compute the target KL divergence as sigmoid(activation) * the KL budget remaining to the agent for the episode. So the activation controls what fraction of the remaining KL budget for the episode to use on the very next token. At initialization, this fraction comes to 1/16. The KL budget remaining starts as the total episode KL budget (of course), and is decreased by $\log(\text{policy}(\text{action})/\text{basepolicy}(\text{action}))$ with each action. The other outputs are interpreted as logits and are added to the base policy logits. Calling this resulting distribution $a$, and the base policy distribution $b$, we find an $\alpha \in [0, 1]$ such that $\text{KL}(\alpha a + (1 - \alpha)b || b)$ equals the target KL, if possible. If we cannot achieve a sufficiently high KL divergence, we set $\alpha = 1$. The output policy is $\alpha a + (1 - \alpha)b$. We add any squared error (target KL − achieved KL)$^2$ to the loss function to encourage the network to output logits that allow further control by the neuron controlling the KL target.

In the forward pass, our custom PyTorch operation does binary search the calculate $\alpha$ in the interval $[0, 1]$. The backward pass uses implicit differentiation, assuming we have found exactly the right $\alpha$—there is no need to differentiate backward through the binary search, which would be unstable.

## F.3    PPO

We use the following hyperparameters for PPO. We do not use a generalized advantage estimate.

| | |
|---|---|
| Training timesteps | 6 million |
| Update frequency | 1 / 64 episodes |
| Training epochs / update | 8 |
| Training batch size | $2^{13}$ |
| Epsilon clip | 0.1 |
| Entropy coefficient | 1e-4 |
| Max gradient norm | 0.1 |
| Actor learning rate | 2e-5 |
| Critic learning rate | 1e-4 |

A higher entropy coefficient is unnecessary given the KL constraint to the base policy. Over the first 3 million timesteps of training, we slowly increase the per-episode KL budget from 0 to its final value. We increase this at a linear schedule each time we update the network.

When we re-train for a longer episode length (256 tokens to 512 tokens), we train for 3 million steps, plenty to reach apparent convergence.

### F.4 PARALLELISM

We use threading to run 64 agent-environment-loops in "parallel". When we would need to send a transcript of length $l$ to be processed by the Mixtral model, we wait until all 64 agent-environment-loops need to send a transcript of length $l$, and then they are batched and evaluated together in parallel on the GPU. The result might needed by either the agent or the environment, and we use the python `asyncio` library to manage this. Doing just that step in parallel is enough for substantial speedup.

### F.5 RESOURCE USAGE

We ran our experiments on two A100-SXM4-80GBs. Training for 9 million timesteps took approximately 90 hours. Our seven training runs (one of which was stopped after 6 million timesteps) took about 25 days, all told. (We ran the experiments two or three at a time). The full research project required much more compute, since finding good hyperparameters for PPO is never straightforward, especially when we were attempting to achieve a desired per-episode KL divergence, only with the use of a fixed per-token KL cost; recall that we eventually switched to a policy architecture that allowed direct control of the per-episode KL divergence.

