# OpenReview forum: "RL, but don't do anything I wouldn't do"
_ICLR.cc/2025/Conference — Submitted to ICLR 2025_

### Official Review · Reviewer_CJKk · 2024-10-27

**Soundness:** 3
**Presentation:** 2
**Contribution:** 3
**Rating:** 6
**Confidence:** 2

**Summary:**

The central point in this paper is that regularizing a policy by KL(policy || base policy) for safety reasons, does not guarantee much safety if all we know about base policy is that KL(safe policy || base policy) is small. This is relevant as many RLHF works use KL regularization as a safety mechanism. The paper shows an interesting theoretical result (Theorem 1) that shows that by minimally affecting the KL distance, a policy can be fine tuned to maximize an arbitrary reward function. The result builds on algorithmic information theory, and looks at a policy that only changes behavior to maximizing reward after some event has occurred, and the idea is to bound how much more complex this policy needs to be (identify the event + compute max reward policy).
The second part of the paper tries to connect this theory to an experiment, where a base conversation agent was fine tuned using RL to maximize the sentiment of the other person’s response under some KL constraint. The agent learned to output empty response, which give neutral sentiment, showing that while KL was regularized, an unwanted performance was obtained.
The third part shows that a specific alternative to KL regularization that is not computationally tractable can in principle avoid the alignment problems outlined in the first part.

**Strengths:**

Disclaimer: my expertise is RL, and I was not familiar with algorithmic information theory before reading the paper.

1. The paper is interesting and thought provoking! I found the connection between KL regularization for RLHF and algorithmic information theory very interesting, and in general I enjoyed reading the paper.
2. The insight that KL regularization cannot guarantee that utility is not optimized (Theorem 1) is important, as this technique is common in RLHF, and alignment is of high interest to a large part of the ICLR community.
3. The experiments, although more of an illustration than an actual result, are interesting and not trivial.

**Weaknesses:**

The biggest weakness is in connecting the theoretical assumptions and results in the paper to a practical meaning.
1. What is the meaning of selecting the prior according to Solomonoff Induction (Lines 141-149)? Why should we expect the RLHF to be related to this prior?
2. In theorem 1, for the bound to be small, we need K(U_m), K(E), and K(v \xi) to be small. I did not understand how we can bound these terms, or why should they be small in practice. Could the authors provide some simple examples that demonstrate the consequences of Theorem 1? The authors discuss the theorem in length, but I found the discussion too vague.
3. In the experiments, a critical factor was that neutral sentiment give reward (0.5) and is easy to obtain. This is a nice demonstration of the point of the paper (high reward, but bad policy). Still, this seems a bit “engineered”, and would have been easy to fix by not giving reward for neutral sentiment. I wonder how the results would look like without reward for neutral sentiment.

Detailed comments:

Line 67: what does “open mindedness” mean here exactly? I found this phrase hard to parse.

Line 77-78: this sentence is very confusing to read, consider revising it to clarify your point. What exactly do these empirical results show?

Line 166: in Definition 2 - for a deterministic environment, can we remove the max over observations? If so, Think that adding this would clarify the motivation for this definition. Also, was this definition considered in prior work?

Line 205: why is K(U_m) small / bounded?

Line 236: “if the RL agent just waits for an unprecedented event with small K(E)” - wouldn’t such an event be very unlikely? Can you say something *in expectation* (or high probability)?

Line 240-249: I found this paragraph very vague and confusing. Why is k the amount of training? Does K(E) necessarily grow with k (is there a formal statement)? 	Can you translate the second half of the paragraph with months of life, etc., into a formal statement?

Line 251: what exactly is the definition of “simplest unprecedented event”?

Line 253-256: I don’t see the connection between algorithmic information theory and rare road conditions (what is the complexity of a road exactly?). Can you provide a more rigorous connection between this paragraph and the previous theorem?

Line 257-259: What if we used Jensen-Shannon Divergence? My guess is that it should fix the problem outlined in Proposition 1.

Line 284: why do you add the feature activations to the agent?

Line 290-303: The KL you refer to here is for single actions, not the KL in definition 2, right? Can you elaborate on the different KL terms you use?

**Questions:**

In addition to the above, there’s something I don’t understand about Theorem 1.
Consider the following example:

There are only two actions, A and B, and no observations. The reward for action A is 0, and the reward for action B is 1. The utility is the average reward (so max is 1).
The imitative policy chooses A with probability 1-epsilon, and therefore, its utility is epsilon. The optimal utility policy always chooses B, and has value V* = 1.
Now, for a policy to obtain reward p at some time step, it needs to choose B with probability p. Then, at that time step, for small epsilon, the KL(policy || imitative) ~ -ln(epsilon)*p > p [using a simple approximation].
In this case, for the policy to obtain high utility, it must also exhibit high KL from the imitative policy.

How does this result reconcile with the explanation for Theorem 1, which claims that “there are policies with near-optimal utility with little KL divergence to an imitative policy”? What am I missing here?

---

> ### Author Response · Authors · 2024-11-20
> **Reply to main comments**
>
> > The paper is interesting and thought provoking! I found the connection between KL regularization for RLHF and algorithmic information theory very interesting, and in general I enjoyed reading the paper.
>
> Thank you!
>
> > What is the meaning of selecting the prior according to Solomonoff Induction (Lines 141-149)? Why should we expect the RLHF to be related to this prior?
>
> It’s the base model that we are expecting to have some connection to this prior. (An improved Figure 1 may help clarify this). We have added a discussion of the connection between Solomonoff Induction and realistic predictive models to the beginning of Section 5. Please let us know what you think. We do not mean to assert that the connection is definite.
>
> > In theorem 1, for the bound to be small, we need K(U_m), K(E), and K(v \xi) to be small. I did not understand how we can bound these terms, or why should they be small in practice. Could the authors provide some simple examples that demonstrate the consequences of Theorem 1? The authors discuss the theorem in length, but I found the discussion too vague.
>
> We have hopefully made this section of the text clearer. After the proof outline, we give a very general example where K(U_m) is small: an agent trained with reinforcement learning, and we have added a bit more explanation. In that setting, the U_m is simply a function that sums some of its arguments. And since the result holds for all E and for all v less than the optimal value, we can note that it holds in particular in cases where K(E) and K(v \xi) are both small. Proposition 2 is a result about the frequency of events for which K(E) is small.
>
> > In the experiments, a critical factor was that neutral sentiment give reward (0.5) and is easy to obtain. This is a nice demonstration of the point of the paper (high reward, but bad policy). Still, this seems a bit “engineered”, and would have been easy to fix by not giving reward for neutral sentiment. I wonder how the results would look like without reward for neutral sentiment.
>
> For what it’s worth, we didn’t anticipate this behavior, although as is often the case with reward hacking, we feel that we should have anticipated it. Our impression is that any given loophole in a reward function is usually easy to fix (at least loopholes that current systems are capable of discovering), but there are just too many possible loopholes to fix them all. Ultimately, this experiment still addresses the question at stake: for a reward function that is positively correlated with good outcomes on the state distribution induced by the base policy, how useful is KL regularization for stopping reward hacking? We can study this question successfully regardless of whether there is another way to stop reward hacking in the particular setting that we are experimenting with. What your comment gets at is that this paper doesn’t convincingly demonstrate that fixing loopholes in reward functions is hard. We agree we do not demonstrate this, but we direct the reader to other literature on that question. And we agree it would be nice to study what the results would look like without reward for neutral sentiment.
>
> For some of the detailed comments, we do not respond here; we have just fixed them in the paper. And for some we respond in a separate comment below.
>
> > there’s something I don’t understand about Theorem 1. Consider the following example … The imitative policy chooses A with probability 1-epsilon … In this case, for the policy to obtain high utility, it must also exhibit high KL from the imitative policy. How does this result reconcile with the explanation for Theorem 1, which claims that “there are policies with near-optimal utility with little KL divergence to an imitative policy”.
>
> First note the KL divergence is only small when a simple unprecedented event occurs. The way to reconcile this is that a Bayesian imitative base policy will not behave as you described in such cases. When faced with a simple unprecedented event, it won't assign probability 1-epsilon to one action. Instead, it will recognize the out-of-distribution situation and hedge its predictions.

---

> ### Author Response · Authors · 2024-11-20
> **Reply to some detailed comments**
>
> For some of the detailed comments, we do not respond here; we have just fixed them in the paper.
>
> > what does “open mindedness” mean here exactly?
>
> It means “being reluctant to rule out hypotheses a priori” or “using a prior with broad support”. An explanation now accompanies the first appearance of closed-/open-minded.
>
> > in Definition 2 - for a deterministic environment, can we remove the max over observations? If so, Think that adding this would clarify the motivation for this definition. Also, was this definition considered in prior work?
>
> Yes we can; we have added this comment. The definition was not considered in prior work to our knowledge, but it is formally just another way to write many KL constraints (no matter what observations are observed) in the space of one equation.
>
> > “if the RL agent just waits for an unprecedented event with small K(E)” - wouldn’t such an event be very unlikely? Can you say something in expectation (or high probability)?
>
> It can be any unprecedented E; the agent doesn’t need to wait for a particular one. This now reads: “if the RL agent just waits for a timestep where there exists an event E with small K(E) that occurs then for the first time, it could then execute an optimal or near-optimal policy. A result below establishes the feasibility of waiting for such a timestep.”
>
> > What if we used Jensen-Shannon Divergence? My guess is that it should fix the problem outlined in Proposition 1.
>
> When $\beta$ is trained to imitate $\tau$, small $KL(\tau || \beta)$ is typically all we can expect, not small $KL(\beta || \tau)$, and therefore not small $JS(\tau || \beta)$. So we cannot combine small $JS(\pi || \beta)$ with small $JS(\tau || \beta)$ to get small $JS(\pi || \tau)$, because we cannot achieve small $JS(\tau || \beta)$ in the first place.
>
> > why do you add the feature activations to the agent?
>
> It’s just easier to do efficient and stable training when the text comprising the state is pre-processed into features. This training regime is equivalent to having the agent observe text, while fixing the parameters of the agent’s policy to match the base policy on the initial layers.
>
> > The KL you refer to here is for single actions, not the KL in definition 2, right? Can you elaborate on the different KL terms you use?
>
> We mention in the previous section that “for autoregressive models, the lifetime KL divergence is equal to the expectation of the sum of the per-timestep KL divergences”.

---

> > ### Comment · Reviewer_CJKk · 2024-11-27
> > **thanks**
> >
> > Thanks for your response and paper edit. I don't have any more questions.
> >
> > Note: I believe there's a typo on line 225.

---

### Official Review · Reviewer_4v2F · 2024-10-29

**Soundness:** 2
**Presentation:** 1
**Contribution:** 3
**Rating:** 3
**Confidence:** 4

**Summary:**

In current llm training, it is common to force the RL policy to be close to a base policy learned by imitating a demonstration policy (the paper calls it trusted policy) from data, using KL regularization. The problem that this paper points out is that current regularization approach can not keep the RL policy close to the demonstration policy.

The paper first notes a simple fact that, for three policies B, D, R,  even if KL(D || B) and KL(R ||B) are both small, KL(R || D) can be infinitely large. This implies that KL(RL policy || demonstration policy) can be large, even if KL(demonstration policy || base policy) and KL(RL policy || base policy) are small. Therefore, RL policy and demonstration policy are not close to each other with KL regularization.

The paper then provides a result showing that there are near-optimal policies with a small KL divergence from a base policy. And given that reward models are not accurate and RL policies typically exploit the weaknesses of the reward models, near-optimal policies are bad policies. This suggests that to keep RL policies not nearly optimal (and therefore bad), the KL regularization needs to be very strong. Further, the same result shows that more imitation learning only slowly increases the KL divergence. The paper then performed an empirical study to verify this theoretical result.

The paper finally shows that for a particular existing imitation learning algorithm of the base policy, which can actively ask for help when facing uncertainty, the above problem is avoided. However, this algorithm is not tractable.

**Strengths:**

The studied problem is closely related to large language model training, which is a popular topic currently.

The results of the paper are novel, non-trivial, and explains certain observation found in large language model training.

The paper shows the authors understanding of the root cause of the problem.

**Weaknesses:**

My main criticism is the quality of the writing of this paper. The paper reads like the flow of the authors' thoughts, instead of an academic paper. The poor writing makes it hard to evaluate the paper's contributions.

Examples:

1. Paragraph right after proposition 2. "Developers of self-driving cars are learning the hard way that this bit of algorithmic information theory has practical analogs: Even with enormous datasets, unprecedented road conditions occur all the time. These results suggest that if we intend to use an imitation learner as a base policy for regularizing a goal-directed agent, we should not strive to approximate ideal Bayesian imitation."

I don't know why the two sentences should appear in the same paragraph. How are they related to each other? And why is this paragraph immediate after proposition 2?

2. The constant d is a small one corresponding to how much code it takes to implement a search tree, Bayes’ rule, and a few if statements.

The paper didn't mention the search tree, Bayes’ rule, and if statements up to this point. Why talk about them?

3. "So unless we use a fairly tight lifetime KL constraint, if the RL agent just waits for an unprecedented event with small K(E), it could then execute an optimal or near-optimal policy, even if that catastrophically thwarts human control, regardless of the content of the base model’s training data, even if the humans that the base model imitates would never, ever behave that way."

I don't understand this sentence. Does such a long sentence effectively convey your ideas?

4. "We say an action is Vξ,U -optimal if it maximizes the associated Q value"

What is Q value?

5. "Let’s consider the case where it is acting in the real world, and maximal reward could be attained by thwarting our control and intervening in its own reward, setting it to a maximal value for all successive timesteps."

What does this mean?

6. "The utility function, simply summing rewards, has an extremely short program length."

Why?

**Questions:**

1. Theorem 2: why is \pi_c^{TVD} function of a_t and x_{<2t}? According to your definition, it is not a function.
2. "As we increase the amount of training k, the Bayesian imitative base model ξ becomes a closer approximation to the humans generating the actions a<k"
Why is k related to the amount of training?
3. From Definition 1, \nu stands for environment transition probability. So \xi should also be. Then why consider the KL divergence between \pi and \xi in Theorem 1?

---

> ### Author Response · Authors · 2024-11-20
>
> > The studied problem is closely related to large language model training, which is a popular topic currently. The results of the paper are novel, non-trivial, and explains certain observation found in large language model training. The paper shows the authors understanding of the root cause of the problem.
>
> Thank you!
>
> > My main criticism is the quality of the writing of this paper. The paper reads like the flow of the authors' thoughts, instead of an academic paper. The poor writing makes it hard to evaluate the paper's contributions.
>
> We appreciate that you provided examples; we have worked to improve the writing there. New text is displayed in teal in our updated draft. I think one issue is that other readers felt they needed more examples and intuitive explanations interspersed through the formal presentation, but if we have done that inelegantly, that may have contributed to your sense of disorganization.
>
> We do not want to assume that your examples are exhaustive. We reviewed the paper’s paragraph structure, and we reviewed the placement of our commentary, intuitive explanations, and instructive examples. This led us to reorganize the middle of Section 4. We hope these edits end up satisfying you enough to increase your score somewhat. Finally, given your summary and your questions, we think you did succeed in evaluating the paper’s contributions. If indeed the presentation style wasn’t such an obstacle to your understanding (notwithstanding several unclear sentences), we hope you might be swayed by other reviewers’ evaluation of our presentation, which ranged from 2-4, and perhaps decrease your confidence in your negative evaluation.
>
> > 1. Theorem 2: why is \pi_c^{TVD} function of a_t and x_{<2t}? According to your definition, it is not a function.
>
> The definition of $\pi_c^{TVD}$ is the argmax over policies of the value, subject to a constraint on the value. (See line 262). So $\pi_c^{TVD}$ is a policy, which takes inputs such as $a_t$ and $x_{<2t}$. Our impression is that it is common in RL to write arg$\max_{\pi}$ as shorthand for arg$\max_{\pi \in \Pi}$, and $\max_{a}$ as shorthand for $\max_{a \in A}$, and so on. But if it would help, we can write it out more thoroughly.
>
> > 2. "As we increase the amount of training k, the Bayesian imitative base model ξ becomes a closer approximation to the humans generating the actions a<k" Why is k related to the amount of training?
>
> The way one “trains” a Bayes’ mixture model like $\xi$ is simply to condition it on the training data. We note that Theorem 1 is of interest when $t > k$, and $\xi$ is conditioned on $x_{<2t}$ in the KL constraint in the theorem, and $x_{<2t}$ includes the actions $a_{<k}$. So the larger $k$ is, the more data $\xi$ is trained on when it is being used for regularizing the RL policy.
>
> > 3. From Definition 1, \nu stands for environment transition probability. So \xi should also be. Then why consider the KL divergence between \pi and \xi in Theorem 1?
>
> The second paragraph of Section 3 defines $\nu: \mathcal{X}^* \times \mathcal{X} \to [0, 1]$ as a probability semi-distribution over every element in the history, actions and observations. In Definition 1, we only use $\nu$ to sample observations, and so we called it an “environment”. We apologize for that; that was ill-done on our part. That seems to be the only place we called $\nu$ an environment, and we now have fixed it. Like $\nu$, $\xi$ is a probability semi-distribution over every element in the history. However, the KL divergence between $\pi$ and $\xi$ (defined in Definition 2) only considers their divergence on action probabilities.

---

> ### Comment · Reviewer_4v2F · 2024-12-03
> **Follow-up questions**
>
> The paper writing improves a bit. However, the discussion on the Theorem 1 is still vague. I have four questions regarding this theorem.
>
> 1. I wonder why when an unprecedented simple event happens, there exists a policy that is close to \xi and also near-optimal, given that the environment emits observations according to \xi.
> 2. Any requirement on how x<2t is generated?
> 3. It looks like \xi can generate both observations and actions, conditioned on the history. So it plays both the role of a policy and the environment. In this case, why do you call it a Bayesian imitator? An imitator usually imitates a policy instead of "imitating" the environment.
> 4. How small can K(vξ(x<2t)) be? I would imagine that ξ(x<2t) is a very small number when t is large. Assume that the reward is sparse and small, then v is also small. In this case, wouldn't K(vξ(x<2t)) be large?

---

> ### Author Response · Authors · 2024-12-03
> **Reply to follow-up questions**
>
> 1. We are considering an agent that uses $\xi$ to model the environment and uses $\xi$ to model the demonstrator. (This is vaguely like having a single network with different output heads for an actor and a critic, just in the sense that one model can do multiple things). When $\xi$ outputs actions, it is predicting what the demonstrator would do; when it outputs observations, it is predicting what the environment would emit.
>
> We motivate why the theorem holds at multiple points in the text. First in the introduction: "a Bayesian imitation learner with a rich prior should be especially reluctant to rule out simple behaviors from the demonstrator in novel settings." $\xi$ is a Bayes' mixture that considers every program as a possible generator of the data. Next in the proof outline, we see that $\xi$ puts meaningful weight on models that switch their behavior to near-optimal behavior after a simple unprecedented event. This is the technical substantiation of the assertion made in the introduction (and later in the text).
>
> 2. None! That may be the most remarkable thing about the theorem. The way we say this in the text is that the theorem holds "regardless of how safe the demonstrator’s policy is". The way that the demonstrator's policy has any effect on the agent's behavior is via its effect on x_{<2t}, so the fact that there are no assumptions about x_{<2t} means the theorem is independent of the nature of the demonstrator.
>
> 3. When we ask $\xi$ to generate actions, it acts as a Bayesian imitator, and that's what we're talking about when we call it a Bayesian imitator.
>
> 4. $vξ(x_{<2t})$ can be very small for the reasons you give, but that does not mean $K(vξ(x_{<2t}))$ becomes large. $K(2^{-2^{100}})$ is very small, because a short programs outputs that number.

---

### Official Review · Reviewer_s47a · 2024-11-03

**Soundness:** 3
**Presentation:** 4
**Contribution:** 4
**Rating:** 8
**Confidence:** 2

**Summary:**

This paper studies the problem of KL-constrained RL, that underpins modern LLM technology, from an algorithmic information theory lens. Their main result suggests that the KL penalty used to avoid the RL fine-tuning process deviating too much from the pre-trained, predictive policy can result in very bad policies that maximize the proxy reward, have small KL, and learn unsafe, undesired behaviors.
The authors formulate the problem by modeling the base/predictive policy as a Bayesian predictive model that is an approximation to the real policy that we do not have access to. Their main result suggests that RL finetuning would make the policy converge to very simple policies that are reward maximizing and have small KL penalty but that diverge from the real policy that we would like to get.

**Strengths:**

1. The paper is very well written: The ideas behind the theory are easy to follow and the authors argue compellingly why this might be a good model of the KL-constrained RL problem
2. The paper is very relevant to modern problems. The paper sheds light on a possible explanation of the performance of RL fine-tuning of imitation policies and they suggest a possible avenue to solve the problem.
3. The empirical evidence is compelling and useful to further understand the theory. The authors use interesting LLM fine-tuning experiments to show how simple rewards can be optimized with simple, policies that have small KL values but converge to policies that are qualitatively far from the desired policies (and the base policy)

**Weaknesses:**

The argument made by the authors is certainly compelling and the empirical evidence seems to suggest that the theory applies. I wonder if algorithmic information theory arguments fit the real-world problem. I guess that there are hypotheses that neural networks approximate such algorithmic theoretical simple programs, however we do not know enough about it. This might be the subject of future research, however, I'd like the authors to comment more on this.

**Questions:**

1. If we are expected to devolve to very simple policies when doing the RL finetuning, why is it that we can tune the process such that we get the performance of modern LLMs?
2. If over-optimizing our proxy reward function causes these struggles, what, you might say, must be the RL objective for this problem if maximizing return is not good?

---

> ### Author Response · Authors · 2024-11-20
>
> > The paper is very well written: The ideas behind the theory are easy to follow and the authors argue compellingly why this might be a good model of the KL-constrained RL problem
>
> Thank you!
>
> > The argument made by the authors is certainly compelling and the empirical evidence seems to suggest that the theory applies. I wonder if algorithmic information theory arguments fit the real-world problem. I guess that there are hypotheses that neural networks approximate such algorithmic theoretical simple programs, however we do not know enough about it. This might be the subject of future research, however, I'd like the authors to comment more on this.
>
> We have added a discussion of why we think so at the beginning of Section 5, but we do not wish to overstate our case by presenting it as indisputable. Please let us know what you think.
>
> > If we are expected to devolve to very simple policies when doing the RL finetuning, why is it that we can tune the process such that we get the performance of modern LLMs?
>
> The fewer loopholes there are in a reward function, the more advanced an RL algorithm needs to be to find them. And today’s RL algorithms appear to be such that we can write reward functions that they struggle to crack (unless they are permitted a significant KL budget). Another contributing factor is that modern RL-finetuned LLMs have a very short time horizon; and it is easier to make hard-to-game reward functions when the RL agent has limited time to game them.
>
> > If over-optimizing our proxy reward function causes these struggles, what, you might say, must be the RL objective for this problem if maximizing return is not good?
>
> Unfortunately, we think there are settings where we cannot construct an RL objective that avoids these struggles. But it could be okay to start with a flawed RL objective but then constrain the RL policy as outlined in Section 6, and optimize that constrained objective.

---

> > ### Comment · Reviewer_s47a · 2024-11-26
> >
> > Thank you for your answers and the added extended discussions in the paper.
> > I will keep my score and confidence the same, as I still believe this is an interesting work.

---

### Official Review · Reviewer_vbPE · 2024-11-04

**Soundness:** 3
**Presentation:** 3
**Contribution:** 3
**Rating:** 8
**Confidence:** 2

**Summary:**

The paper investigates the effectiveness of KL regularization as a safety paradigm for RL agents. The authors, using the formalism of algorithmic information theory, show that if one uses KL to regularize the behaviour of RL agent, one would eventually need to continually impose a very tight KL constraint. In fact, they show in Theorem 1 that there exists near-optimal policies with little KL divergence to an imitative policy. In the work this is showed that this is not only singular to KL, but to other regularizations such as TVD.

The paper further proposes empirical experiments to validate their theoretical results. Specifically, they do so in a scenario when an LLM simulating a teacher has as an objective to maximize positive sentiments from the students. The empirical results show that by increasing the KL budget, the agent learns to remain silent. Together with other results, these 'imperefect' experiments support the theoretically results proved.

Finally, the author propose to overcome the problems highlighted by proposing a theoretical alternative, which relies on the RL agent asking for help to the human demonstrator if at hand. Given the intractability of this approach, it was not possible for the authors to empirically validate this approach.

**Strengths:**

The paper proposes novel, and technically solid ideas. In many points along the paper the authors provide additional intuitions on the significance of the results, which have been useful to grasp the subtleties of the theoretical results. The continual links between the claims made, the results showed and the implications that they have made the paper easy to follow and clear to understand, although this is not precisely my area of expertise. Also good to include a proof outline as now, e.g. for Theorem 1.

I am also positively surprised that the paper provides empirical experiments to validate the theory. By reading the first part of the paper I would have imagined this to be difficult or lead to an oversimplification, but I am actually satisfied by these results. I still believe some additional experiments could be done to improve this ( see questions).

More importantly, I think the claims are correctly calibrated - given the technical ( and almost speculative, in a positive sense) nature of the paper, it would have been easy to fall in overestimating the impact and significance of the results. The authors did a very good job in clearly stating the limitations of their work, both for the theoretical and empirical part.

**Weaknesses:**

The main weakness of the paper regards the assumptions made in Section 4 (see also questions below). Most of them are sound, but I believe part of the community would not necessarily agree with this. I think it could be beneficial for the paper to have additional results where some of the assumptions are weaken, and investigate which theoretical results would derive in tis case. For example, what happens if we drop the assumption that all overoptimized policies lead to unsafe behaviours?

Another point is that it is not completely clear to me that future powerful AI systems will in fact respect the assumptions made in the paper. While I agree that current AI systems are not powerful enough to fully support the theoretical results, it would be insightful to add some discussion regarding why the authors think that this paradigm will necessary be relevant for more powerful AI systems.

**Questions:**

- Could Theorem 1 be summarized as saying "once an event E which can be described with a small K(E) happens, given the bound given the RL agent may be able to exploit this to perform a (near-)optimal policy while remaining in the KL constraint. Thus, given that maximally optimizing the true reward leads to unsafe behaviour, this may lead to catastrophic behaviour". If that is the case, I have two questions related to this: (1) Is there any constraint on which type this event E needs to be? In other words, does this hold for all unprecedented events with small K(E), or for a subset of them which may be related with unsafe behaviour? (2) If we drop the assumption that overoptimization necessarily leads to catastrophic behaviours but let's say happens with probability $p$ once overoptimizing, how can Theorem 1 be extended? Is then the probability of having catastrophic behaviour just proportional to $p$, or we are ensured that in that case the KL would be enough?

- I suggest making Figure 1 more explanatory, for example by adding the notation relevant to the component nearby each box. I believe this would be possible if making the image larger, and would be a good reference for the readers to appropriately understand the settings and go back to it if needed while reading the paper.

- What results would be obtained if instead of doing KL(rl policy|| base/imitative policy) one uses KL(base/imitative policy||rl policy)?

- Any explanation for the bump on the left Figure 3, around the right side of the x-axis?

- "when the agent discovers a sufficiently high-reward strategy, the fixed KL penalty becomes swamped and ignored, and if the KL penalty is increased to a level where it can stop that, the agent never gets off the ground". I believe it would be interesting to see results where this is the case, especially looking at which is the threshold for the KL penalty to exhibit one behaviour or the other. Did you observe a

- It seems a bit arbitrary to run experiments solely with a budget of 10 or 20. It would be good to provide additional experiments where the budget is varied, and have a plot where on the x-axis there are a few budget data points, and on the y-axis a metric that aggregates the fraction of responses empty.

I will be keen to raise my score if the above questions and doubts are addressed appropriately.

---

> ### Author Response · Authors · 2024-11-20
>
> > The main weakness of the paper regards the assumptions made in Section 4 … For example, what happens if we drop the assumption that all overoptimized policies lead to unsafe behaviours?
>
> One shouldn't consult this paper when designing an agent if any of the following are true: A) all highly-optimized policies are safe (as is the case for a chess-playing agent); B) some highly-optimized policies are safe, and some are not, but one has a method for steering the agent toward the safe policies; or C) one does not have such a method, but one simply has a high risk-tolerance and is willing to try vanilla RL. In a setting where our assumption is inappropriate, we genuinely have little advice to offer. But if there are unsafe highly-optimized policies, one lacks a method to avoid them, and one isn’t willing to take the risk, then we provide helpful guidance. We don’t quite assume that all overoptimized policies lead to unsafe behaviors; we say “assume a setting where if [π is near-optimal], then π is considered unacceptably risky.”
>
> Unfortunately, we do not see a way to model this risk precisely. You suggest a setting where catastrophic behavior “happens with probability p once overoptimizing”. But it's unclear how to probabilistically model which policy an RL algorithm will adopt, so we can’t easily work backwards from the final probability p(catastrophe) to a more sophisticated result. Instead, we integrate over this uncertainty and consider settings where the risk of an unacceptable outcome is unacceptable.
>
> > … it would be insightful to add some discussion regarding why the authors think that this paradigm will necessarily be relevant for more powerful AI systems
>
> I don’t think we would say that this paradigm will necessarily be relevant. We think it might well be relevant. We have added a discussion of why we think so at the beginning of Section 5, but we do not wish to overstate our case by presenting it as indisputable. Please let us know what you think.
>
> > Is there any constraint on which type this event E needs to be? In other words, does this hold for all unprecedented events with small K(E), or for a subset of them which may be related with unsafe behaviour?
>
> No constraint at all! It holds for all events E, whether K(E) is small or not, and whether E has anything to do with unsafe behaviour. But if K(E) is not small, then the resulting bound is weak, because the KL divergence is shown to be less than a large number.
>
> > making Figure 1 more explanatory
>
> Done.
>
> > What results would be obtained if instead of doing KL(rl policy|| base/imitative policy) one uses KL(base/imitative policy||rl policy)?
>
> At the very beginning of the episode, flip some coins. If they all come up heads, follow the optimal policy; else follow the base policy. This has small KL(π || β), and if the optimal policy is bad, then this is strictly worse than doing no RL at all and following the base policy. Like with TVD regularization, this regularization does not appear to help even if we had a perfect imitation of a trusted policy.
>
> > Any explanation for the bump on the left Figure 3, around the right side of the x-axis?
>
> This is the only place on the plot where the error bars are large enough to see, because the sample size is small. Not many episodes have that many empty responses.
>
> > "when the agent discovers a sufficiently high-reward strategy, the fixed KL penalty becomes swamped and ignored, and if the KL penalty is increased to a level where it can stop that, the agent never gets off the ground". I believe it would be interesting to see results where this is the case
>
> We would be a bit embarrassed to present an analysis of this threshold, because a) there is extreme variation depending on the random seed, and b) to some extent it would just be a study of our own failings; in industry, they seem to find a way to avoid this. We found our own way to avoid it, so we want to present that method and why we developed it, but we’re not sure we’re prepared to present high quality results on this question.
>
> > It would be good to provide additional experiments where the budget is varied, and have a plot where on the x-axis there are a few budget data points, and on the y-axis a metric that aggregates the fraction of responses empty.
>
> On the topic of budget… this is a very expensive plot to produce given the difficulty of RL-finetuning language models, and we’re not sure whether it would be worth the dollar cost. The question that that plot would be an answer to is probably better answered by Gao, et al. (2022) Figure 1. We explored a KL budget of 8, and there was only one training run where it learned to be silent, even after we attempted to reproduce that. Presumably better or more training would have produced that behavior more reliably. It would therefore be incorrect to say that we can expect empty responses to be rare for a KL-budget 8 agent, but it would be very expensive to produce a high-quality estimate of their frequency.

---

> > ### Comment · Reviewer_vbPE · 2024-11-25
> >
> > Thanks for your answers.
> >
> > The introduction to 5 is interesting, and agree that claims should not be overstated. The position of the authors is much clearer now, at least to me. I would suggest also inserting what came after in your answer "One shouldn't consult this paper.." in the paper somewhere as well - I find it useful again to understand the claims you were making ( albeit I am not super familiar with algorithmic information theory).
> >
> > The other answers to my more conceptual questions also clarified my doubts
> >
> > Regarding the experiments: while I may understand the costs argument for my second request, I think having negative results (my 1st request) should not cause any 'embarassement'. These experiments may be expensive, and I also do not think the main contributions of this paper are empirical, hence this does not hinder my judgment of the paper. However, I  suggest the authors to include such a result in an appendix in case they find the time, I think it would be something interesting to have
> >
> > I will raise my score to 8, and I will keep my confidence at 2.

---

### Meta-Review · Area_Chair_YQHe · 2024-12-23

**Metareview:**

This paper examines the limitations of using KL regularization between the agent policy and the imitator policy to enforce a small KL divergence between the agent policy and the demonstrator policy. I have the following major concerns with this submission:


- Poor Writing Quality: I strongly agree with Reviewer 4v2F that the overall writing quality is well below the acceptance standard. The paper contains numerous vague, confusing or notationally dense statements. A complete rewrite would significantly improve clarity and make the work more accessible to the general ML audience.

- Weak Connection Between Theory and Experiments: As Reviewer CJKk pointed out, the connection between the theoretical results and the experimental evidence is weak. It is easy to construct a contrived experimental setup where a model collapses despite adhering to a small KL budget. As a result, it is unclear how much the experiments meaningfully support the theoretical claims.




- Confusing Motivation for the Main Theoretical Results: The main theoretical claim states that having both KL(A, I) and KL(I, D) small does not guarantee that KL(A, D) will also be small, where A, I, and D denote the agent policy, the imitator policy, and the demonstrator policy, respectively. However, while this is true for KL divergence, it is well-known that small KL(A, I) and KL(I, D) do imply that theTV distance between A and D is small. In the context of language models, two models with a small TV distance will generate outputs that are very similar in a probabilistic sense, making TV distance a meaningful metric. To counter this, the paper argues that TV distance being small might be insufficient because it does not prevent the agent policy from taking actions outside the demonstrator policy’s support with a small positive probability. However, recovering the support of the demonstrator policy from finite samples is fundamentally impossible from an information-theoretic perspective. Unless we are extremely conservative in terms of generalization, it’s in general impossible to avoid going out of the support of the demonstrator policy. In essence, the paper critiques TV distance for failing to address a problem that is more or less impossible.

**Additional Comments On Reviewer Discussion:**

This submission received polarized reviews. On one hand, two reviewers provided very positive scores, albeit with relatively low confidence, and their concerns appeared to be adequately addressed during the rebuttal phase. On the other hand, one reviewer found the paper difficult to parse and expressed strong criticism of its presentation. While the authors made efforts to improve the writing during the rebuttal phase, after reviewing the revised version myself, I believe it still falls well below the standard required for publication. Therefore, I recommend rejection.

---

> ### Public Comment · ~Michael_K._Cohen1 · 2025-02-08
> **Reply to meta review**
>
> We never got the chance to reply to the new technical concerns raised in this meta-review. (Some supreme court members prefer not to advance completely new arguments in their decisions that have not been advanced by either side in court, because they have not had the benefit of hearing adversarial responses.)
>
> > It is easy to construct a contrived experimental setup where a model collapses despite adhering to a small KL budget. As a result, it is unclear how much the experiments meaningfully support the theoretical claims.
>
> Yes it easy. And *why* is it easy? This paper offers an explanation, and we are not aware of any other explanations in the literature. If one thinks about it, one should find it somewhat surprising that it is easy to observe model collapse with a small KL budget. The field is used to papers where the only point of an experiment is to demonstrate algorithmic ingenuity. But what does "easiness" have to do with inability to "support" a claim? (In fact, this setup wasn't contrived; this was the first setup we attempted, so our task was even easier than this comment suggests!)
>
> > In the context of language models, two models with a small TV distance will generate outputs that are very similar in a probabilistic sense, making TV distance a meaningful metric. To counter this, the paper argues that TV distance being small might be insufficient because it does not prevent the agent policy from taking actions outside the demonstrator policy’s support with a small positive probability.
>
> When the demonstrator policy puts low probability on a course of action, that can be because it is extremely bad. So the ratio of probabilities P_imitator / P_demonstrator is more relevant for safety.
>
> > In essence, the paper critiques TV distance for failing to address a problem that is more or less impossible.
>
> Please see Section 6 for a solution to this problem.

---

### Decision · Program_Chairs · 2025-01-22

Reject